# GRAPH MLP-MIXER

## ABSTRACT

Graph Neural Networks (GNNs) have shown great potential in the field of graph representation learning. Standard GNNs define a local message-passing mechanism which propagates information over the whole graph domain by stacking multiple layers. This paradigm suffers from two major limitations, over-squashing and poor long-range dependencies, that can be solved using global attention but significantly increases the computational cost to quadratic complexity. In this work, we consider an alternative approach to overcome these structural limitations while keeping a low complexity cost. Motivated by the recent MLP-Mixer architecture introduced in computer vision, we propose to generalize this network to graphs. This GNN model, namely Graph MLP-Mixer, can make long-range connections without over-squashing or high complexity due to the mixer layer applied to the graph patches extracted from the original graph. As a result, this architecture exhibits promising results when comparing standard GNNs vs. Graph MLP-Mixers on benchmark graph datasets.

## 1 BACKGROUND AND MOTIVATION

In this section, we review the main classes of GNNs with their advantages and their limitations. Then, we introduce the ViT/MLP-Mixer architectures from computer vision which have motivated us to design a new graph network architecture.

**Message-Passing GNNs (MP-GNNs).** GNNs have become the standard learning architectures for graphs based on their flexibility to work with complex data domains s.a. recommendation (Monti et al., 2017; van den Berg et al., 2018), chemistry (Duvenaud et al., 2015; Gilmer et al., 2017), physics (Cranmer et al., 2019; Bapst et al., 2020), transportation (Derrow-Pinion et al., 2021), vision (Han et al., 2022), NLP (Wu et al., 2021a), knowledge graphs (Schlichtkrull et al., 2018), drug design (Stokes et al., 2020; Gaudelet et al., 2020) and medical domain (Li et al., 2020b; 2021). Most GNNs are designed to have two core components. First, a structural message-passing mechanism s.a. Defferrard et al. (2016); Kipf & Welling (2017); Hamilton et al. (2017); Monti et al. (2017); Bresson & Laurent (2017); Gilmer et al. (2017); Veličković et al. (2018) that computes node representations by aggregating the local 1-hop neighborhood information. Second, a stack of $L$ layers that aggregates $L$-hop neighborhood nodes to increase the expressivity of the network and transmit information between nodes that are $L$-hops apart.

**Weisfeiler-Leman GNNs (WL-GNNs).** One of the major limitations of MP-GNNs is their inability to distinguish (simple) non-isomorphic graphs. This limited expressivity can be formally analyzed with the Weisfeiler-Leman graph isomorphism test (Weisfeiler & Leman, 1968), as first proposed in Xu et al. (2019); Morris et al. (2019). Later on, Maron et al. (2018) introduced a general class of $k$-order WL-GNNs that can be proved to universally represent any class of $k$-WL graphs (Maron et al., 2019; Chen et al., 2019). But to achieve such expressivity, this class of GNNs requires using $k$-tuples of nodes with memory and speed complexities of $O(N^k)$, with $N$ being the number of nodes and $k \geq 3$. Although the complexity can be reduced to $O(N^2)$ and $O(N^3)$ respectively (Maron et al., 2019; Chen et al., 2019; Azizian & Lelarge, 2020), it is still computationally costly compared to the linear complexity $O(E)$ of MP-GNNs, which often reduces to $O(N)$ for real-world graphs that exhibit sparse structures s.a. molecules, knowledge graphs, transportation networks, gene regulatory networks, to name a few. In order to reduce memory and speed complexities of WL-GNNs while keeping high expressivity, several works have focused on designing graph networks from their sub-structures s.a. sub-graph isomorphism (Bouritsas et al., 2022), sub-graph routing mechanism (Alsentzer et al., 2020), cellular WL sub-graphs (Bodnar et al., 2021), expressive sub-

graphs (Bevilacqua et al., 2021; Frasca et al., 2022), rooted sub-graphs (Zhang & Li, 2021) and k-hop egonet sub-graphs (Zhao et al., 2021a).

**Graph Positional Encoding (PE).** Another aspect of the limited expressivity of GNNs is their inability to recognize simple graph structures s.a. cycles or cliques, which are often present in molecules and social graphs (Chen et al., 2020). We can consider $k$-order WL-GNNs with value $k$ to be the length of cycle/clique, but with high complexity $O(N^k)$. An alternative approach is to add positional encoding to the graph nodes. It was proved in Murphy et al. (2019); Loukas (2020) that unique and equivariant PE increases the representation power of any MP-GNN while keeping the linear complexity. This theoretical result was applied with great empirical success by Murphy et al. (2019) with index PE, Dwivedi et al. (2020); Dwivedi & Bresson (2021); Kreuzer et al. (2021); Lim et al. (2022) with Laplacian eigenvectors and Li et al. (2020a); Dwivedi et al. (2021) with k-step Random Walk. All these graph PEs lead to GNNs strictly more powerful than the 1-WL test, which seems to be enough expressivity in practice (Zopf, 2022). However, none of the PE proposed for graphs can provide a global position of the nodes that is unique, equivariant and distance sensitive. This is due to the fact that a canonical positioning of nodes does not exist for arbitrary graphs, as there is no notion of up, down, left and right on graphs. For example, any embedding coordinate system like graph Laplacian eigenvectors (Belkin & Niyogi, 2003) can flip up-down directions, right-left directions, and would still be a valid PE. This introduces ambiguities for the GNNs that require to (learn to) be invariant with respect to the graph or PE symmetries. A well-known example is given by the eigenvectors: there exist $2^k$ number of possible sign flips for $k$ eigenvectors that require to be learned by the network.

**Over-Squashing.** Standard MP-GNNs require $L$ layers to propagate the information from one node to their $L$-hop neighborhood. This implies that the receptive field size for GNNs can grow exponentially, for example with $O(2^L)$ for binary tree graphs. This causes over-squashing; information from the exponentially-growing receptive field is compressed into a fixed-length vector by the aggregation mechanism (Alon & Yahav, 2020; Topping et al., 2022). Consequences of over-squashing are overfitting and poor long-range node interactions as relevant information cannot travel without being disturbed. Over-squashing is well-known since recurrent neural networks (Hochreiter & Schmidhuber, 1997), which have led to the development of the (self- and cross-)attention mechanisms for the translation task (Bahdanau et al., 2014; Vaswani et al., 2017) first, and then for more general natural language processing (NLP) tasks (Devlin et al., 2018; Brown et al., 2020). Transformer architectures are the most elaborated networks that leverage attention. Attention is a simple but powerful mechanism that solves over-squashing and long-range dependencies by making "everything connected to everything" but it also requires to trade linear complexity for quadratic complexity. Inspired by the great successes of Transformers in NLP and computer vision (CV), several works have proposed to generalize the transformer architecture for graphs , achieving competitive or superior performance against standard MP-GNNs. We highlight the most recent research works in the next paragraph.

**Graph Transformers.** GraphTransformers (Dwivedi & Bresson, 2021) generalize Transformers to graphs, with graph Laplacian eigenvectors as node PE, and incorporating graph structure into the permutation-invariant attention function. SAN and LSPE (Kreuzer et al., 2021; Dwivedi et al., 2021) further improve with PE learned from Laplacian and random walk operators. GraphiT (Mialon et al., 2021) encodes relative PE derived from diffusion kernels into the attention mechanism. GraphTrans (Wu et al., 2021b) and SAT (Chen et al., 2022) add Transformers on the top of standard GNN layers. Graphormer (Ying et al., 2021) introduce three structural encodings, with great success on large molecular benchmarks. GPS (Rampášek et al., 2022) categorizes the different types of PE and puts forward a hybrid MPNN+Transformer architecture. We refer to Min et al. (2022) for an overview of graph-structured Transformers. Generally, most Graph Transformer architectures address the problems of over-squashing and limited long-range dependencies in GNNs but they also increase significantly the complexity from $O(E)$ to $O(N^2)$, resulting in a computational bottleneck.

**ViT and MLP-Mixer.** Transformers have gained remarkable success in CV and NLP, most notably with architectures like ViT (Dosovitskiy et al., 2020) and BERT (Devlin et al., 2018). The success of transformers has been long attributed to the attention mechanism (Vaswani et al., 2017), which is able to model long-range dependencies as it does not suffer from over-squashing. But recently, this prominent line of networks has been challenged by more cost efficient alternatives. A novel family of models based on the MLP-Mixer introduced by Tolstikhin et al. (2021) has emerged and gained recognition for its simplicity and its efficient implementation. Overall, MLP-Mixer replaces the attention module with multi-layer perceptrons (MLPs) which are also not affected by over-squashing

and poor long-range dependencies. The original architecture is simple (Tolstikhin et al., 2021); it takes image patches (or tokens) as inputs, encodes them with a linear layer (equivalent to a convolutional layer over the image patches), and updates their representations with a series of feed-forward layers applied alternatively to image patches (or tokens) and features. The follow-up variants investigate different mixing operations, such as ResMLP (Touvron et al., 2021), gMLP (Liu et al., 2021), and DynaMixer (Wang et al., 2022). These plain networks can perform competitively with state-of-the-art vision Transformers, which tends to indicate that attention is not the only important inductive bias, but other elements like the general architecture of Transformers with patch embedding, residual connection and layer normalization, and carefully-curated data augmentation techniques seem to play essential roles as well (Yu et al., 2022).

**Main Objective.** Motivated by the MLP-Mixer introduced in CV, our goal is to investigate a generalization of this architecture from grids to graphs. The motivation is clear; MLP-Mixer offers a low-cost alternative to ViT for images, avoiding the quadratic complexity of the attention mechanism while keeping long-range interactions. We wish to transfer these advantages to GNNs. Our contributions are as follows.

- We identify the key challenges to generalize MLP-Mixer from images to graphs.
- We design a new GNN, namely Graph MLP-Mixer, that is not limited by over-squashing and poor long-distance dependencies while keeping the linear complexity of MP-GNNs.
- We report extensive experiments to analyze the proposed GNN architecture with several datasets from the Benchmarking GNNs (Dwivedi et al., 2020) and the Open Graph Benchmark (OGB) (Hu et al., 2020).
- Our approach forms a bridge between CV, NLP and graphs under a unified architecture, that can potentially benefit cross-over domain collaborations to design better networks.

## 2 GENERALIZATION CHALLENGES

In the following, we list the main questions when adapting MLP-Mixer from images to graphs.

**(1) How to define and extract graph patches/tokens?** One notable geometrical property that distinguishes graph-structured data from regular structured data, such as images and sequences, is that there does not exist in general a canonical grid to embed graphs. As shown in Table 1, images are supported by a regular lattice, which can be easily split into multiple grid-like patches (also referred to as tokens) of the same size via fast pixel reordering. However, graph data is irregular: the number of nodes and edges in different graphs is typically different. Hence, graphs cannot be uniformly divided into similar patches across all examples in the dataset. Finally, the extraction process for graph patches cannot be uniquely defined given the lack of canonical graph embedding. This raises the questions of how we identify meaningful graph tokens, and quickly extract them.

**(2) How to encode graph patches into a vectorial representation?** Since images can be reshaped into patches of the same size, they can be linearly encoded with an MLP, or equivalently with a convolutional layer with kernel size and stride values equal to the patch size. However, graph patches are not all the same size: they have variable topological structure with different number of nodes, edges and connectivity. Another important difference is the absence of a unique node ordering for graphs, which constrains the process to be invariant to node re-indexing for generalization purposes. In summary, we need a process that can transform graph patches into a fixed-length vectorial representation for arbitrary subgraph structures while being permutation invariant. GNNs are naturally designed to perform such transformations, and as such will be used to encode graph patches.

**(3) How to preserve positional information for nodes and graph patches?** As shown in Table 1, image patches in the sequence have implicit positions since image data is always ordered the same way due to its unique embedding in the Euclidean space. For instance, the image patch at the upper-left corner is always the first one in the sequence and the image patch at the bottom-right corner is the last one. On this basis, the token mixing operation of the MLP-Mixer is able to fuse the same patch information. However, graphs are naturally not-aligned and the set of graph patches are therefore unordered. We face a similar issue when we consider the positions of nodes within each graph patch. In images, the pixels in each patch are always ordered the same way; in contrast, nodes in graph tokens are naturally unordered. Thus, how do we preserve local and global positional consistency for graph patches, and nodes in each patch?

| | Images | Graphs |
|---|---|---|
| |  |  |
| Input | Regular grid
Same data resolution
(Height, Width) | Irregular domain
Variable data structure
(# Nodes and # Edges) |
| Patch Extraction | Via pixel reordering
Non-overlapping patches
Same patches at each epoch | Via graph clustering algorithm
Overlapping patches
Different patches at each epoch |
| Patch Encoder | Same patch resolution
(Patch Height, Patch Width)
MLP (equivalently CNN) | Variable patch structure
(# Nodes and # Edges)
GNN (e.g. GCN, GAT, GT) |
| Positional Information | Implicitly ordered
(No need for explicit PE) | No universal ordering
Node PE for patch encoder
Patch PE for token mixer |
| MLP-Mixer | Channel mixer
Token mixer | Channel mixer
Token mixer with patch PE |

Table 1: Differences between MLP-Mixer components for images and graphs.

**(4) How to reduce over-fitting for Graph MLP-Mixer?** Most MLP-variants (Tolstikhin et al., 2021; Touvron et al., 2021; Wang et al., 2022) first pre-train on large-scale datasets, and then fine-tune on downstream tasks, coupled with a rich set of data augmentation and regularization techniques, e.g. cropping, random horizontal flipping, RandAugment (Cubuk et al., 2020), mixup (Zhang et al., 2017), etc. While data augmentation has drawn much attention in CV and NLP, graph data augmentation methods are not yet as effective, albeit interest and works on this topic (Zhao et al., 2021b). Variable number of nodes, edges and connectivity make graph augmentation challenging. Thus, how do we augment graph-structured data given this nature of graphs?

We summarize the differences between standard MLP-Mixer and Graph MLP-Mixer in Table 1.

## 3 PROPOSED ARCHITECTURE

### 3.1 OVERVIEW

The basic architecture is illustrated in Figure 1. The goal of this section is to detail the choices we made to implement each component of the architecture. On the whole, these choices lead to a simple framework that provides good practice performance.

**Notation.** Let $G = (\mathcal{V}, \mathcal{E})$ be a graph with $\mathcal{V}$ being the set of nodes and $\mathcal{E}$ the set of edges. The graph has $N = |\mathcal{V}|$ nodes and $E = |\mathcal{E}|$ edges. The connectivity of the graph is represented by the adjacency matrix $A \in \mathbb{R}^{N \times N}$. The node features of node $i$ are denoted by $h_i$, while the features for an edge between nodes $i$ and $j$ are indicated by $e_{ij}$. Let $\{\mathcal{V}_1, ..., \mathcal{V}_P\}$ be the nodes partition, $P$ be the pre-defined number of patches, and $G_i = (\mathcal{V}_i, \mathcal{E}_i)$ be the induced subgraph of $G$ with all the nodes in $\mathcal{V}_i$ and all the edges whose endpoints belong to $\mathcal{V}_i$. Let $h_G$ be the graph-level vectorial representation and $y_G$ be the graph-level target, which can be a discrete variable for graph classification problem, or a scalar for graph regression task.

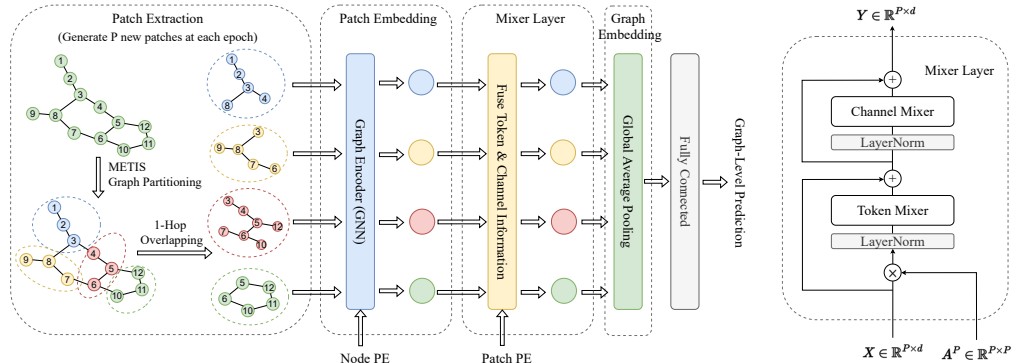

Figure 1: The basic architecture of the proposed Graph MLP-Mixer. Graph MLP-Mixer consists of a patch extraction module, a patch embedding module, a mixer layer, a global average pooling, and a classifier head. The patch extraction module partitions graphs into overlapping patches. The patch embedding module transforms these graph patches into corresponding token representations, which are fed into a sequence of mixer layers to generate the output tokens. A global average pooling layer followed by a fully-connected layer is finally used for prediction. Each Mixer Layer is a residual network that alternates between a Token Mixer applied to all patches, and a Channel Mixer applied to each patch independently.

## 3.2 Patch Extraction

When generalizing MLP-Mixer to graphs, the first step is to extract patches. This extraction is straightforward for images. Indeed, all image data $x \in \mathbb{R}^{H \times W \times C}$ are defined on a regular grid with the same fixed resolution $(H, W)$, where $H$ and $W$ are respectively the height and the width, and $C$ is the number of channels. Hence, all images can be easily reshaped into a sequence of flattened patches $x_p \in \mathbb{R}^{P \times (R^2 C)}$, where $(R, R)$ is the resolution of each image patch, and $P = HW/R^2$ is the resulting number of patches, see Table 1.

Unlike images with fixed resolution, extracting graph patches is more challenging. Generally, graphs have different sizes, i.e. number of nodes, and therefore cannot be uniformly divided like image data. Additionally, meaningful sub-graphs must be identified in the sense that nodes and edges composing a patch must share similar semantic or information, s.a. a community of friends sharing biking interest in a social network. As such, a graph patch extraction process must satisfy the following conditions: (1) The same extraction algorithm can be applied to any arbitrary graph, (2) The nodes in the sub-graph patch must be more closely connected than for those outside the patch, and (3) The extraction complexity must be fast, that is at most linear w.r.t. the number of edges, i.e. $O(E)$.

Graph partitioning algorithms have been studied for decades (Buluç et al., 2016) given their importance in identifying meaningful clusters. Mathematically, graph partitioning is known to be NP-hard (Chung, 1997). Approximations are thus required. A graph clustering algorithm with one of the best trade-off accuracy and speed is METIS (Karypis & Kumar, 1998), which partitions a graph into a pre-defined number of clusters/patches such that the number of within-cluster links is much higher than between-cluster links in order to better capture good community structure. For these fine properties, we select METIS as our graph patch extraction algorithm.

However, METIS is limited to finding non-overlapping clusters, as visualized in Figure 1. In this example, METIS partitions the graph into four non-overlapping parts, i.e. $\{1, 2, 3\}, \{4, 5, 6\}, \{7, 8, 9\}$ and $\{10, 11, 12\}$, resulting in 5 edge cuts. Unlike images, extracting non-overlapping patches could imply losing important edge information, i.e. the cutting edges, and thus decreasing the predictive performance, as we will observe experimentally. To overcome this issue and to retain all original edges, we allow graph patches to overlap with each other. For example in Figure 1, if the source and destination nodes of an edge are not in the same patch, we assign both nodes to the patches they belong to. As such, node 3 and node 4 are in two different patches, here the blue and red one, but are connected with each other. After our overlapping adjustment, these two nodes belong to both the blue and red patches. This practice is equivalent to expanding the graph patches to the one-hop neighbourhood of all nodes in that patch. Formally, METIS is first applied to partition a graph into

$P$ non-overlapping patches: $\{\mathcal{V}_1, ..., \mathcal{V}_P\}$ such that $\mathcal{V} = \mathcal{V}_1 \cup ... \cup \mathcal{V}_P$ and $\mathcal{V}_i \cap \mathcal{V}_j = \emptyset$, $\forall i \neq j$. Then, patches are expanded to their one-hop neighbourhood in order to preserve the information of between-patch links and make use of all graph edges: $\mathcal{V}_i \leftarrow \mathcal{V}_i \cup \{ \mathcal{N}_1(j) \mid j \in \mathcal{V}_i \}$, where $\mathcal{N}_k(j)$ defines the $k$-hop neighbourhood of node $j$.

### 3.3 PATCH ENCODER

For images, patch encoding can be done with a simple linear transformation given the fixed resolution of all image patches. This operation is fast and well-defined. For graphs, the patch encoder network must be able to handle complex data structure such as invariance to index permutation, heterogeneous neighborhood, variable patch sizes, convolution on graphs, and expressive to differentiate graph isomorphisms. As a result, the graph patch encoder is a GNN, whose architecture is designed to best transform a graph token $G_p$ into a fixed-size representation $x_{G_p} \in \mathbb{R}^d$ into 3 steps.

**Step 1. Raw node and edge linear embedding.** The input node features $\alpha_i \in \mathbb{R}^{d_n \times 1}$ and edge features $\beta_{ij} \in \mathbb{R}^{d_e \times 1}$ are linearly projected into $d$-dimensional hidden features:

$$h_i^0 = U^0 \alpha_i + u^0 \in \mathbb{R}^d; \quad e_{ij}^0 = V^0 \beta_{ij} + v^0 \in \mathbb{R}^d \tag{1}$$

where $U^0 \in \mathbb{R}^{d \times d_n}$, $V^0 \in \mathbb{R}^{d \times d_e}$ and $u^0, v^0 \in \mathbb{R}^d$ are learnable parameters.

**Step 2. Graph convolutional layers with (favorite) GNN.** We apply a series of $L$ convolutions to improve the patch representation of node and edge features:

$$\begin{aligned}
h_i^{\ell+1} &= f_h(h_i^\ell, \{h_j^\ell | j \in \mathcal{N}(i)\}, e_{ij}^\ell) \quad h_i^{\ell+1}, h_i^\ell \in \mathbb{R}^d, \\
e_{ij}^{\ell+1} &= f_e(h_i^\ell, h_j^\ell, e_{ij}^\ell) \quad e_{ij}^{\ell+1}, e_{ij}^\ell \in \mathbb{R}^d,
\end{aligned} \tag{2}$$

where $\ell$ is the layer index, functions $f_h$ and $f_e$ (with learnable parameters) define the specific GNN architecture, and $\mathcal{N}(i)$ is the neighborhood of the node $i$.

**Step 3. Pooling and readout.** The final step produces a fixed-size vector representation by mean pooling all node vectors in $G_p$ and applying a small MLP to get the patch embedding $x_{G_p}$.

The patch encoder is a GNN, and thus has the same potential limitations of over-squashing and poor long-range dependencies. However, these problems become prominent only for large graphs. But for small patch graphs, such problems do not really exist (or are negligible). Indeed, in practice, the mean number of nodes and the mean diameter for graph patches are around 3.2 and 1.8 respectively for molecular datasets and around 12.0 and 2.7 for image datasets, see Table 9.

### 3.4 POSITIONAL INFORMATION

Regular grids offer a natural implicit arrangement for the sequence of image patches and for the pixels inside the image patches. However, such ordering of nodes and patches do not exist for general graphs. This lack of positional information reduces the expressivity of the network. Hence, we use explicitly one absolute PE for the patch nodes and one relative PE for the graph patches. **Node PE.** Input node features in Eq 1 are augmented with $p_i \in \mathbb{R}^K$:

$$h_i^0 = T^0 p_i + U^0 \alpha_i + u^0 \in \mathbb{R}^d, \tag{3}$$

where $T^0 \in \mathbb{R}^{d \times K}$ is a learnable matrix. The benefits of different PEs are dataset dependent. We follow the strategy in Rampášek et al. (2022) that uses random-walk structural encoding (RWSE) (Dwivedi et al., 2021) for molecular data and Laplacian eigenvectors encodings (Dwivedi et al., 2020) for image superpixels. Since Laplacian eigenvectors are defined up to sign flips, the sign of the eigenvectors is randomly flipped during training.

**Patch PE.** Relative positional information between the graph patches can be computed from the original graph adjacency matrix $A \in \mathbb{R}^{N \times N}$ and the clusters $\{\mathcal{V}_1, ..., \mathcal{V}_P\}$ extracted by METIS in Section 3.2. Specifically, we capture relative positional information via the 'coarsened adjacency matrix' $A^P \in \mathbb{R}^{P \times P}$ over the patch graphs:

$$A_{ij}^P = |\mathcal{V}_i \cap \mathcal{V}_j| = \text{Cut}(\mathcal{V}_i, \mathcal{V}_j), \tag{4}$$

where $\text{Cut}(\mathcal{V}_i, \mathcal{V}_j) = \sum_{k \in \mathcal{V}_i} \sum_{l \in \mathcal{V}_j} A_{kl}$ is the standard graph cut operator which counts the number of connecting edges between cluster $\mathcal{V}_i$ and cluster $\mathcal{V}_j$. We observe that matrix $A^P$ is sparse as it

only connects patches that are neighbors on the original graph. This can cause poor long-distance interactions. To avoid this situation, we can simply smooth out the adjacency matrix $A^P$ with any graph diffusion process. In this work, we select the n-step random walk diffusion process:

$$A_{\mathcal{D}}^P = \left(D^{-1} A^P\right)^n \in \mathbb{R}^{P \times P} \tag{5}$$

## 3.5 Mixer Layer

For images, the original mixer layer in Tolstikhin et al. (2021) is a simple network that alternates channel and token mixing steps. The token mixing step is performed over the token dimension, while the channel mixing step is carried out over the channel dimension. These two interleaved steps enable information fusion among tokens and channels. The simplicity of the mixer layer has been of great importance to understand that the self-attention mechanism in ViT is not the only critical component to get good performance on visual classification tasks. This has also led to a significant reduction in computational cost with little or no sacrifice in performance. Indeed, the self-attention mechanism in ViT requires $O(P^2)$ memory and $O(P^2)$ computation, while the mixer layer in MLP-Mixer needs $O(P)$ memory and $O(P)$ computation.

We modify the original mixer layer to introduce positional information between graph tokens. Let $X \in \mathbb{R}^{P \times d}$ be the patch embedding $\{x_{G_1}, ..., x_{G_P}\}$. The graph mixer layer can be expressed as

$$
\begin{aligned}
U &= X + (W_2 \, \sigma(W_1 \, \text{LayerNorm}(A_{\mathcal{D}}^P X))) \in \mathbb{R}^{P \times d} &&\text{Token mixer,} \\
Y &= U + (W_4 \, \sigma(W_3 \, \text{LayerNorm}(U)^T))^T \in \mathbb{R}^{P \times d} &&\text{Channel mixer,}
\end{aligned}
\tag{6}
$$

where $A_{\mathcal{D}}^P \in \mathbb{R}^{P \times P}$ is the patch PE from Eq.5, $\sigma$ is a GELU nonlinearity (Hendrycks & Gimpel, 2016), LayerNorm($\cdot$) is layer normalization (Ba et al., 2016), and matrices $W_1 \in \mathbb{R}^{d_s \times P}, W_2 \in \mathbb{R}^{P \times d_s}, W_3 \in \mathbb{R}^{d_c \times d}, W_4 \in \mathbb{R}^{d \times d_c}$, where $d_s$ and $d_c$ are the tunable hidden widths in the token-mixing and channel-mixing MLPs, and are set following Tolstikhin et al. (2021).

We generate the final graph-level representation by mean pooling all the non-empty patches:

$$h_G = \sum_p m_p \cdot x_{G_p} / \sum_p m_p \ \in \mathbb{R}^d, \tag{7}$$

where $m_p$ is a binary variable with value 1 for non-empty patches and value 0 for empty patches (since graphs have variable sizes, and thus small graphs can produce empty patches). Finally, we apply a small MLP to get the graph-level target:

$$y_G = \text{MLP}(h_G) \tag{8}$$

## 3.6 Data augmentation

MLP-Mixer architectures are known to be strong over-fitters (Liu et al., 2021). In order to reduce this effect, we perform data augmentation of graph patches. At each epoch, we randomly drop a few edges before running METIS partitioning, to produce more diverse partitions. This data augmentation process is very fast as METIS graph clustering only amounts to a small portion of the data preparation time, therefore adding little extra cost during the training processes.

## 4 Experiments

**Graph Benchmark Datasets.** We conduct extensive experiments to investigate the proposed method. From the Benchmarking GNNs (Dwivedi et al., 2020), we test on ZINC, MNIST and CI-FAR10. From the open graph benchmark (OGB) (Hu et al., 2020), we test on MolHIV, MolTOX21, and MolPCBA. Summary statistics of datasets are reported in Table 6 and Appendix A.1.

**Extraction Step: Study of # patches.** We observe in Figure 5 that increasing the number of patches generally improves performance, which is consistent with computer vision (Dosovitskiy et al., 2020; Tolstikhin et al., 2021). We set the number of patches to 16/32 by default. The resulting graph patches are of small size; they typically contain 3-12 nodes with a value diameter of 2-3, see Table 9.

**Extraction Step: Study of k-hop extension.** In Figure 2 and Table 10, we observe a clear performance increase when graph patches are overlapping with each other (0-hop vs 1-hop), which is

Figure 2: Effect of the $k$-hop extension. We expand graph patches extracted by METIS to the $k$-hop neighbourhood of all nodes in that patch. 0-hop means non-overlapping patches without extension.

| Model | ZINC | MNIST | CIFAR10 | MolTOX21 | MolHIV | MolPCBA | Peptide-func | Peptide-struct |
|---|---|---|---|---|---|---|---|---|
| | MAE ↓ | Accuracy ↑ | Accuracy ↑ | ROCAUC ↑ | ROCAUC ↑ | Avg. Precision ↑ | Avg. Precision ↑ | MAE ↓ |
| GCN | 0.1956 ± 0.0030 | 0.9574 ± 0.0016 | 0.5876 ± 0.0028 | 0.7644 ± 0.0076 | 0.7665 ± 0.0093 | **0.2549 ± 0.0044** | 0.6376 ± 0.0044 | 0.2745 ± 0.0010 |
| GCN-MLP-Mixer | **0.1880 ± 0.0067** | **0.9606 ± 0.0019** | **0.5892 ± 0.0024** | **0.7737 ± 0.0062** | **0.7865 ± 0.0130** | 0.2525 ± 0.0015 | 0.6545 ± 0.0075 | 0.2509 ± 0.0011 |
| GCN-MLP-Mixer* | **0.1352 ± 0.0056** | **0.9773 ± 0.0007** | **0.6579 ± 0.0042** | **0.7824 ± 0.0041** | 0.7849 ± 0.0043 | 0.2486 ± 0.0046 | **0.6736 ± 0.0040** | **0.2488 ± 0.0012** |
| GatedGCN | 0.1565 ± 0.0023 | 0.9815 ± 0.0010 | 0.6703 ± 0.0024 | 0.7659 ± 0.0061 | 0.7673 ± 0.0123 | **0.2721 ± 0.0036** | 0.6452 ± 0.0111 | 0.2812 ± 0.0023 |
| GatedGCN-MLP-Mixer | **0.1540 ± 0.0046** | **0.9823 ± 0.0011** | **0.6804 ± 0.0043** | **0.7783 ± 0.0041** | **0.7824 ± 0.0062** | 0.2669 ± 0.0022 | **0.6636 ± 0.0039** | 0.2519 ± 0.0016 |
| GatedGCN-MLP-Mixer* | **0.1251 ± 0.0080** | **0.9831 ± 0.0010** | **0.6896 ± 0.0021** | **0.7828 ± 0.0050** | **0.7912 ± 0.0133** | 0.2649 ± 0.0015 | **0.6790 ± 0.0073** | **0.2481 ± 0.0013** |
| GINE | 0.1183 ± 0.0027 | 0.9740 ± 0.0013 | 0.6031 ± 0.0030 | 0.7669 ± 0.0052 | 0.7660 ± 0.0156 | **0.2749 ± 0.0021** | 0.6616 ± 0.0091 | 0.2768 ± 0.0014 |
| GINE-MLP-Mixer | **0.1171 ± 0.0029** | **0.9778 ± 0.0014** | **0.6053 ± 0.0082** | **0.7780 ± 0.0079** | **0.7824 ± 0.0131** | 0.2613 ± 0.0015 | 0.6627 ± 0.0106 | 0.2498 ± 0.0028 |
| GINE-MLP-Mixer* | **0.0794 ± 0.0028** | **0.9787 ± 0.0012** | **0.6625 ± 0.0044** | **0.7839 ± 0.0035** | **0.7844 ± 0.0033** | 0.2666 ± 0.0029 | **0.6846 ± 0.0068** | **0.2478 ± 0.0010** |
| GraphTrans | 0.1286 ± 0.0029 | 0.9732 ± 0.0014 | 0.6876 ± 0.0077 | 0.7672 ± 0.0065 | 0.7786 ± 0.0089 | 0.2098 ± 0.0032 | 0.6548 ± 0.0031 | 0.2768 ± 0.0014 |
| GraphTrans-MLP-Mixer | **0.1190 ± 0.0016** | **0.9798 ± 0.0008** | **0.7010 ± 0.0041** | **0.7748 ± 0.0079** | **0.7805 ± 0.0182** | **0.2433 ± 0.0037** | 0.6585 ± 0.0040 | 0.2522 ± 0.0025 |
| GraphTrans-MLP-Mixer* | **0.0831 ± 0.0032** | **0.9827 ± 0.0008** | **0.7023 ± 0.0046** | **0.7827 ± 0.0065** | **0.7823 ± 0.0217** | 0.2214 ± 0.0055 | **0.6814 ± 0.0072** | **0.2483 ± 0.0013** |
| GNN-free MLP-Mixer | 0.1290 ± 0.0045 | 0.9717 ± 0.0008 | 0.5917 ± 0.0009 | 0.7738 ± 0.0048 | 0.7852 ± 0.0131 | 0.2199 ± 0.0061 | – | – |

Table 2: Performance on Benchmarking datasets (Dwivedi et al., 2020), OGB (Hu et al., 2020) and LRGB (Dwivedi et al., 2022). Shown is the mean ± s.d. of 4 runs.

consistent with our intuition that extracting non-overlapping patches implies losing important edge information. We further expand graph patches to their $k$-hop neighbourhood. Performance increases first and then flattens out or begins to decrease when $k = 3$ for ZINC and $k = 1/2$ for MolHIV.

**Encoder Step: GNN-based patch encoder.** We evaluate the effect of various GNN models as patch encoder in Table 2, which includes GCN (Kipf & Welling, 2017), GatedGCN (Bresson & Laurent, 2017), GINE (Hu et al., 2019) and Graph Transformer (Shi et al., 2020). We find that our GNN-MLP-Mixer architecture matches or outperforms existing GNNs across different datasets and patch encoders. These promising results demonstrate the generic nature of the proposed Graph MLP-Mixer architecture which can be applied to any MP-GNNs in practice.

**Encoder Step: GNN-free patch encoder.** We also investigate a GNN-free patch encoder. For each patch, we embed all node and edge features as bags of nodes and edges, then average and readout, where all transformations are based exclusively on MLPs, see Eq.9. Interestingly, this GNN-free MLP-Mixer produces good results (last row of Table 2), which seems to imply that the GNN-encoder is not critical in this architecture. Even more excitingly, further development of Graph MLP-Mixer may not need to use specialized GNN libraries like DGL (Wang et al., 2019) or PyG (Fey & Lenssen, 2019a) to achieve competitive performance with standard MP-GNNs.

**Encoder Step: Updated node encoding.** We consider a more expressive GNN-MLP-Mixer by updating the node representation with the patch representation coming from the mixer layer. We call this improved version GNN-MLP-Mixer* and we present the details in Appendix A.9.

**Positional Information.** We show the effects of two kinds of positional encoding in Figure 3. First, we observe a significant drop in performance when either node PE or patch PE is removed. Besides, it can be observed that the extent of poor performance of models without node PE against using node PE is greater for ZINC than MolHIV. This difference can be explained by the fact that ZINC features are purely atom and bond descriptors whereas MolHIV features consist additional information that is informative of e.g. if an atom is in ring, among others.

**Data Augmentation.** Then proposed data augmentation (DA) corresponds to newly generated graph patches with METIS at each epoch, while no DA means patches are only generated at the initial epoch and then reused during training. Table 3 presents different results. First, it is clear that DA

Figure 3: Effect of positional information. We study the effects of node PE and patch PE by removing one of them in turn from our model while keeping the other components unchanged.

**ZINC**

| Methods | GCN-MLP-Mixer | GatedGCN-MLP-Mixer | GINE-MLP-Mixer | GraphTrans-MLP-Mixer |
|---------|---------------|--------------------|-----------------|-----------------------|
| Ours | 0.1880 | 0.1540 | 0.1190 | 0.1190 |
| w/o node PE | 0.4704 | 0.4484 | 0.3983 | 0.4611 |
| w/o patch PE | 0.1906 | 0.1634 | 0.1193 | 0.1243 |

**MolHIV**

| Methods | GCN-MLP-Mixer | GatedGCN-MLP-Mixer | GINE-MLP-Mixer | GraphTrans-MLP-Mixer |
|---------|---------------|--------------------|-----------------|-----------------------|
| Ours | 0.7865 | 0.7824 | 0.7824 | 0.7805 |
| w/o node PE | 0.7700 | 0.7776 | 0.7786 | 0.7773 |
| w/o patch PE | 0.7731 | 0.7661 | 0.7679 | 0.7631 |

**Peptides-func**

| Methods | GCN-MLP-Mixer | GatedGCN-MLP-Mixer | GINE-MLP-Mixer | GraphTrans-MLP-Mixer |
|---------|---------------|--------------------|-----------------|-----------------------|
| Ours | 0.6736 | 0.6790 | 0.6846 | 0.6814 |
| w/o node PE | 0.6563 | 0.6665 | 0.6801 | 0.6699 |
| w/o patch PE | 0.6650 | 0.6703 | 0.6672 | 0.6642 |

| Dataset | DA | MAE↓ / ROCAUC↑ | Time(S/Epoch) |
|---------|-----|-----------------|----------------|
| ZINC | ✗ | 0.1266 ± 0.0051 | 5.9216 |
|  | ✓ | 0.1171 ± 0.0029 | 5.9922 |
| MolHIV | ✗ | 0.7569 ± 0.0281 | 14.8949 |
|  | ✓ | 0.7824 ± 0.0131 | 15.5889 |

Table 3: Effect of data augmentation (DA): ✗ means no DA and ✓ uses DA.

| Dataset | Model | MAE↓ / ROCAUC↑ |
|---------|-------|-----------------|
| ZINC | GINE-MLP-Mixer* | 0.0794 ± 0.0028 |
|  | GINE-ViT | 0.1060 ± 0.0014 |
| MolHIV | GINE-MLP-Mixer* | 0.7844 ± 0.0033 |
|  | GINE-ViT | 0.7694 ± 0.0158 |

Table 4: Mixer Layer vs. Transformer Layer.

brings an increase in performance. Second, re-generating graph patches only add to a small amount of training time. Full results are reported in Table 12.

| Model | ZINC | MolHIV | MolPCBA | Peptides-func | Peptides-struct |
|-------|------|--------|---------|----------------|------------------|
|  | MAE ↓ | ROCAUC ↑ | ROCAUC ↑ | Avg. Precision ↑ | MAE ↓ |
| GT (Dwivedi & Bresson, 2021) | 0.226 ± 0.014 | – | – | – | – |
| GraphiT (Mialon et al., 2021) | 0.202 ± 0.011 | – | – | – | – |
| Graphormer (Ying et al., 2021) | 0.122 ± 0.006 | – | – | – | – |
| GPS (Rampášek et al., 2022) | 0.070 ± 0.004 | 0.7880 ± 0.0101 | 0.2907 ± 0.0028 | 0.6562 ± 0.0115 | 0.2515 ± 0.0012 |
| SAN (Chen et al., 2022) | 0.139 ± 0.006 | 0.7775 ± 0.0061 | 0.2765 ± 0.0042 | 0.6439 ± 0.0075 | 0.2545 ± 0.0012 |
| GraphTrans (Kreuzer et al., 2021) | – | – | 0.2761 ± 0.0029 | – | – |
| GNN-AK+ (Alsentzer et al., 2020) | 0.080 ± 0.001 | 0.7961 ± 0.0119 | 0.2930 ± 0.0044 | 0.6480 ± 0.0089 | 0.2736 ± 0.0007 |
| SUN (Frasca et al., 2022) | 0.084 ± 0.002 | 0.8003 ± 0.0055 | 0.2616 ± 0.0049 | 0.6730 ± 0.0078 | 0.2498 ± 0.0008 |
| Graph MLP-Mixer* (Ours) | 0.0794 ± 0.003 | 0.7912 ± 0.0133 | 0.2749 ± 0.0021 | **0.6846 ± 0.0068** | **0.2478 ± 0.0010** |

Table 5: Comparison of our best results from Table 2 with the state-of-the-art GTs (missing values from literature are indicated with '-'). For ZINC, all models have approximately ∼ 500k parameters.

**Mixer Layer: MLP vs. Attention.** In Table 4, we replace MLP-Mixer layers with standard Transformer layers while keeping the rest of the components the same. The performance of MLP-Mixer is surprisingly better than the latter despite a lower complexity. Full results are reported in Table 13.

**State-Of-The-Art.** Table 5 presents the SOTA GraphTransformer (GT) models. To ensure fair comparison, we did not include Graphormer (Ying et al., 2021) that achieved top score on MolHIV after pre-training on a large dataset of 3.8M graphs. Overall, we observe that our Graph MLP-Mixer model achieves competitive performance without making use of the fully-connected attention mechanism, solely using low-cost mixer operations. Besides, we are more efficient in model parameters and training time. Full comparison with number of training parameters, memory and training time is provided Table 14 and Table 15 in the appendix.

**Expressivity.** We experimentally show that Graph MLP-Mixer is strictly more powerful than 1-WL, and not less powerful than 3-WL. Although graph PEs s.a. Laplacian eigenvectors (Belkin & Niyogi, 2003) or k-step Random Walk PE (Li et al., 2020a; Dwivedi et al., 2021) cannot guarantee two

graphs are generally isomorphic, it was shown in Dwivedi et al. (2021) that they can distinguish non-isomorphic graphs for which the 1-WL test fails. As a consequence, Graph MLP-Mixer is strictly more powerful than 1-WL. We experimentally validate this property by running Graph MLP-Mixer on the highly symmetric Circulant Skip Link (CSL) dataset of Murphy et al. (2019) in Table 16, which requires GNNs to be strictly more expressive than the 1-WL test to succeed. Besides, Graph MLP-Mixer reaches perfect accuracy on SR25 (Balcilar et al., 2021). The SR25 dataset contains 15 strongly regular graphs with 25 nodes each, where no model less or as powerful as 3-WL test can distinguish the pairs in SR25 dataset.

**Long Range Graph Benchmark (LRGB).** We evaluate our models and compare with the baselines on the LRGB (Dwivedi et al., 2022) with 2 graph-level datasets, *i.e.,* Peptides-func and Peptides-struct, that arguably require long-range information reasoning to achieve strong performance in the given tasks. As shown in Table 2 and Table 5, Graph MLP-Mixer performs significantly better than the baselines, especially on Peptides-struct. The improvement can be explained by the nature of these datasets, which is consistent with the empirical findings in (Dwivedi et al., 2022) that simple instances of local MP-GNNs perform poorly on the proposed datasets. More information is provided in Table 15.

## 5    CONCLUSION

In this work, we have proposed a novel GNN model directly inspired from ViT/MLP-Mixer architectures in computer vision and presented promising results on benchmark graph datasets. Future work will focus on further exploring graph networks with the inductive biases of graph tokens and Transformer-like architectures in order to solve fundamental node and link prediction tasks, and potentially without the need of specialized GNN libraries.

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

# A  Experimental Details

## A.1  Datasets description

**CSL** is a synthetic dataset introduced in Murphy et al. (2019) to test the expressivity of GNNs. CSL has 150 graphs. Each CSL graph is a 4-regular graph with edges connected to form a cycle and containing skip-links between nodes.

**SR25** is another synthetic dataset used to empirically verify the expressive power of Graph MLP-Mixer. SR25 (Balcilar et al., 2021) has 15 strongly regular graphs (3-WL failed) with 25 nodes each. SR25 is translated to a 15 way classification problem with the goal of mapping each graph into a different class.

**ZINC** (Dwivedi et al., 2020) is a subset (12K) of molecular graphs (250K) from a free database of commercially-available compounds (Irwin et al., 2012). These molecular graphs are between 9 and 37 nodes large. Each node represents a heavy atom (28 possible atom types) and each edge represents a bond (3 possible types). The task is to regress a molecular property known as the constrained solubility. The dataset comes with a predefined 10K/1K/1K train/validation/test split.

**MNIST and CIFAR10** (Dwivedi et al., 2020) are derived from classical image classification datasets by constructing an 8 nearest-neighbor graph of SLIC superpixels for each image. The resultant graphs are of sizes 40-75 nodes for MNIST and 85-150 nodes for CIFAR10. The 10-class classification tasks and standard dataset splits follow the original image classification datasets, i.e., for MNIST 55K/5K/10K and for CIFAR10 45K/5K/10K train/validation/test graphs. These datasets are sanity-checks, as we expect most GNNs to perform close to 100% for MNIST and well enough for CIFAR10.

**MolTOX21 and MolHIV** (Hu et al., 2020) are molecular property prediction datasets adopted from the MoleculeNet (Szklarczyk et al., 2019). All the molecules are pre-processed using RDKit (Landrum et al., 2006). Each graph represents a molecule, where nodes are atoms, and edges are chemical bonds. Input node features are 9-dimensional, containing atomic number and chirality, as well as other additional atom features such as formal charge and whether the atom is in the ring or not. The datasets come with a predefined scaffold splits based on their two-dimensional structural frameworks, i.e. for MolTOX21 6K/0.78K/0.78K and for MolHIV 32K/4K/4K train/validation/test.

**MolPCBA** (Hu et al., 2020) is another real-world molecular graph classification benchmark. MOLTOX21 and MolHIV are of small and medium scale with 7.8K and 41.1k graphs respectively, whereas MOLPCBA is of large scale with 437.9K graphs, and applies a similar scaffold spliting procedure. It consists of multiple, extremely skewed (only 1.4% positivity) molecular classification tasks, and employs Average Precision (AP) over them as a metric.

**Peptides-func and Peptides-struct** (Dwivedi et al., 2022) are derived from 15,535 peptides with a total of 2.3 million nodes retrieved from SAT-Pdb (Singh et al., 2016). Both datasets use the same set of graphs but differ in their prediction tasks. These graphs are constructed in such a way that requires long-range interactions (LRI) reasoning to achieve strong performance in a given task. In concrete terms, they are larger graphs: on average 150.94 nodes per graph, and on average 56.99 graph diameter. Thus, they are better suited to benchmarking of graph Transformers or other expressive GNNs that are intended to capture LRI.

**Distributions of the graph sizes.** We plot of the distributions of the graph sizes (i.e. the number of nodes in each data sample) of these datasets in Figure 4.

## A.2  Hyperparameters

We follow the benchmarking protocol introduced in Dwivedi et al. (2020) based on PyTorch (Paszke et al., 2019) and PyG (Fey & Lenssen, 2019b). We use Adam (Kingma & Ba, 2014) optimizer, with the default settings of $\beta_1 = 0.9$, $\beta_2 = 0.999$, and $\epsilon = 1e^{-8}$. The learning rate is reduced by half if the validation loss does not improve after 10 epochs. The training stops at a point when the learning rate reaches to a value of $1 \times 10^{-5}$. We use 4 layers of GNN layers for patch encoder and 4 layers of Mixer layers by default. For benchmarking datasets from Dwivedi et al. (2020), the dropout is set to 0. For OGB datasets from Hu et al. (2020), we tune the dropout $\in \{0, 0.1, 0.2, 0.3, 0.4, 0.5\}$

| Dataset | #Graphs | #Nodes | Avg. #Nodes | Avg. #Edges | Task | Metric |
|---|---|---|---|---|---|---|
| CSL | 150 | 41 | 41 | 164 | 10-class classif. | Accuracy |
| SR25 | 15 | 25 | 25 | 300 | 15-class classif. | Accuracy |
| ZINC | 12,000 | 9-37 | 23.2 | 24.9 | regression | MAE |
| MNIST | 70,000 | 40-75 | 70.6 | 564.5 | 10-class classif. | Accuracy |
| CIFAR10 | 60,000 | 85-150 | 117.6 | 941.1 | 10-class classif. | Accuracy |
| MolTOX21 | 7831 | 1-132 | 18.57 | 38.6 | 12-task classif. | ROCAUC |
| MolHIV | 41,127 | 2-222 | 25.5 | 27.5 | binary classif. | ROCAUC |
| MolPCBA | 437,929 | 1–332 | 26.0 | 28.1 | 28-task classif. | Avg. Precision |
| Peptides-func | 15,535 | 2,344,859 | 150.94 | 307.30 | 10-clas classif. | Avg. Precision |
| Peptides-struct | 15,535 | 2,344,859 | 150.94 | 307.30 | regression | MAE |

Table 6: Summary statistics of datasets used in this study

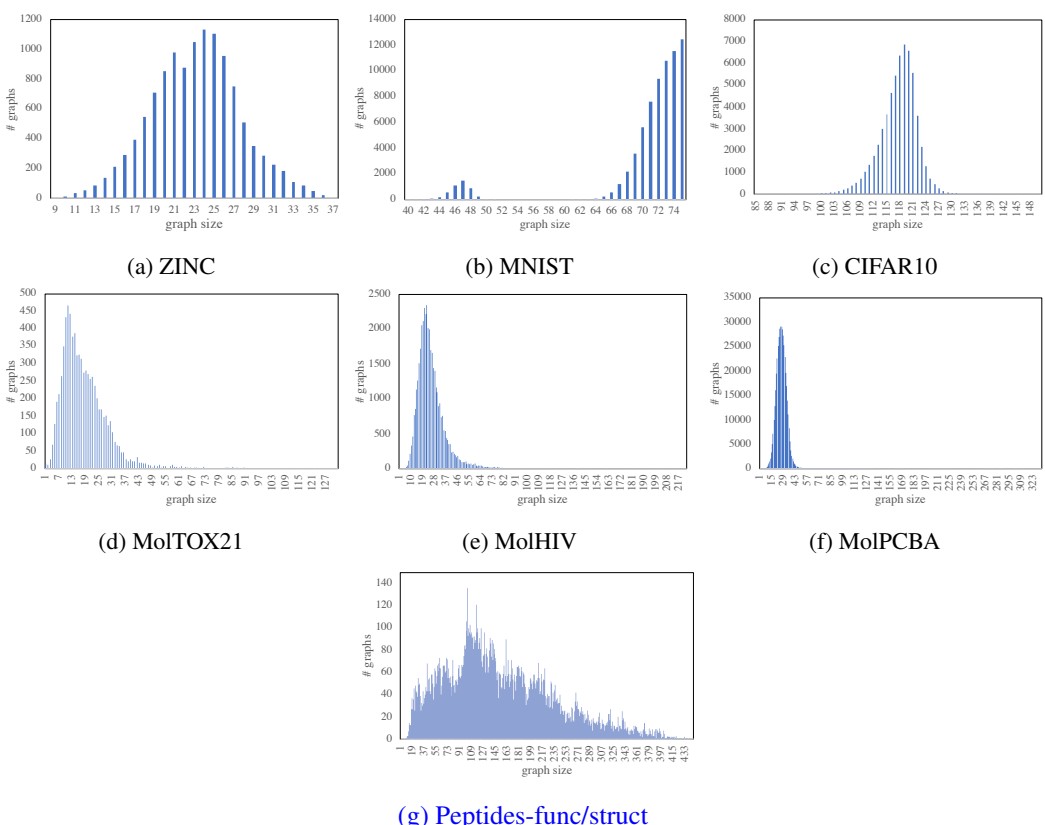

(a) ZINC  (b) MNIST  (c) CIFAR10

(d) MolTOX21  (e) MolHIV  (f) MolPCBA

(g) Peptides-func/struct

Figure 4: Distributions of the graph sizes.

for both the baseline and our methods. We run each experiment 4 times, and report the mean ± s.d. performance. Detailed hyperparameters are provided in Table 7 and Table 8.

## A.3 PATCH EXTRACTION

**Study of # patches.** We observe in Figure 5 that increasing the number of patches generally improves performance.

**Patch Size and Diameter.** We set the number of patches to 16/32 by default. The resulting graph patches are of small size, as shown in Table 9.

| Hyperparameter | ZINC | MNIST | CIFAR10 | CSL | SR25 |
|---|---|---|---|---|---|
| # Patch | 32 | 16 | 16 | 32 | 16 |
| # GNN layer | 4 | 4 | 4 | 4 | 4 |
| # MLP-Mixer layer | 4 | 4 | 4 | 4 | 4 |
| Hidden size | 128 | 128 | 128 | 128 | 128 |
| Learning rate | 0.001 | 0.001 | 0.001 | 5e-4 | 0.002 |
| Batch size | 128 | 128 | 128 | 5 | 15 |
| Node PE | RWSE-20 | LapPE-8 | LapPE-8 | LapPE-20/RWSE-20 | LapPE-8 |

Table 7: Model hyperparameters for four datasets from Dwivedi et al. (2020)

| Hyperparameter | MolTOX21 | MolHIV | MolPCBA |
|---|---|---|---|
| # Patch | 32 | 32 | 32 |
| # GNN layer | 4 | 4 | 4 |
| # MLP-Mixer layer | 4 | 4 | 4 |
| Hidden size | 128 | 128 | 400 |
| Learning rate | 0.001 | 0.001 | 0.005 |
| Batch size | 128 | 128 | 512 |
| Node PE | – | RWSE-16 | RWSE-16 |

Table 8: Model hyper-parameters for three datasets from OGB (Hu et al., 2020)

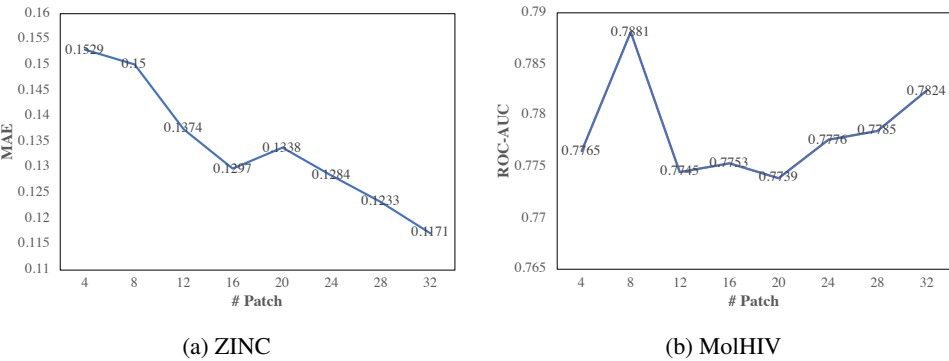

(a) ZINC  (b) MolHIV

Figure 5: Effect of the number of patches.

| Dataset | # Patch | # Node | | | Diameter | | |
|---|---|---|---|---|---|---|---|
| | | Mean | Min | Max | Mean | Min | Max |
| ZINC | 32 | 3.1495 | 2 | 5 | 1.8151 | 2 | 1 |
| MNIST | 16 | 10.85 | 6 | 18 | 2.1240 | 2 | 4 |
| CIFAR10 | 16 | 11.9163 | 7 | 20 | 2.6413 | 2 | 6 |
| MolTox21 | 32 | 3.0272 | 1 | 7 | 1.7293 | 0 | 4 |
| MolHIV | 32 | 3.1822 | 1 | 11 | 1.8362 | 0 | 6 |
| MolPCBA | 32 | 3.1905 | 1 | 12 | 1.8640 | 0 | 9 |
| Peptides-func/struct | 32 | 5.3943 | 1 | 17 | 3.4407 | 1 | 13 |

Table 9: Summary statistics of graph patches for different datasets.

**Effect of k-hop extension.** Table 10 is the full result of Figure 2. For other experiments, we set $k = 1$ by default.

**Study of Graph Partitioning Algorithm.** We select METIS (Karypis & Kumar, 1998) as the graph partitioning algorithm. In table 11, we provide the ablation study for how many benefits

| | # Hop | GCN-MLP-Mixer | GatedGCN-MLP-Mixer | GINE-MLP-Mixer | GraphTrans-MLP-Mixer |
|---|---|---|---|---|---|
| ZINC | 0 | 0.2568 ± 0.0034 | 0.2642 ± 0.0037 | 0.2462 ± 0.0018 | 0.2602 ± 0.0085 |
| | 1 | 0.1880 ± 0.0067 | 0.1540 ± 0.0046 | 0.1171 ± 0.0029 | 0.1190 ± 0.0016 |
| | 2 | **0.1780 ± 0.0041** | 0.1468 ± 0.0016 | 0.1046 ± 0.0039 | 0.1126 ± 0.0036 |
| | 3 | 0.1820 ± 0.0074 | **0.1443 ± 0.0061** | **0.1009 ± 0.0032** | **0.1072 ± 0.0032** |
| | 4 | 0.2022 ± 0.0038 | 0.1473 ± 0.0046 | 0.1072 ± 0.0011 | 0.1174 ± 0.0083 |
| MolHIV | 0 | 0.7736 ± 0.0106 | 0.7706 ± 0.0162 | 0.7636 ± 0.0118 | 0.7684 ± 0.0176 |
| | 1 | 0.7788 ± 0.0191 | **0.7772 ± 0.0118** | 0.7711 ± 0.0124 | **0.7805 ± 0.0182** |
| | 2 | **0.7885 ± 0.0103** | 0.7668 ± 0.0125 | **0.7797 ± 0.0062** | 0.7665 ± 0.0125 |
| | 3 | 0.7716 ± 0.0015 | 0.7586 ± 0.0162 | 0.7734 ± 0.0096 | 0.7425 ± 0.0086 |

Table 10: Study of k-hop neighbour on ZINC and MolHIV, corresponding to Fig. 2

the METIS can provide against random graph partitioning. For random graph partition, nodes are randomly assigned to a pre-defined number of patches. For both METIS and random partition, we re-generate graph patches at each epoch. Table 11 shows that using METIS as the graph partition algorithm consistently gives better performance than random node partition, which corresponds to our intuition that nodes and edges composing a patch should share similar semantic or information. Nevertheless, it is interesting to see that random graph partitioning is still able to achieve reasonable results, which shows that the performance of the model is not solely supported by the quality of the patches.

| Model | ZINC | | MolHIV | |
|---|---|---|---|---|
| | METIS | Random | METIS | Random |
| GCN-MLP-Mixer* | 0.1352 ± 0.0056 | 0.1846 ± 0.0042 | 0.7849 ± 0.0043 | 0.7791 ± 0.0252 |
| GatedGCN-MLP-Mixer* | 0.1251 ± 0.0080 | 0.1619 ± 0.0071 | 0.7912 ± 0.0133 | 0.7721 ± 0.0117 |
| GINE-MLP-Mixer* | 0.0794 ± 0.0028 | 0.1057 ± 0.0048 | 0.7844 ± 0.0033 | 0.7676 ± 0.0101 |
| GraphTrans-MLP-Mixer* | 0.0831 ± 0.0032 | 0.1166 ± 0.0051 | 0.7823 ± 0.0217 | 0.7772 ± 0.0099 |

Table 11: Comparison of METIS vs. random graph partitioning.

### A.4 PATCH ENCODER

**Baselines and GNN-based patch encoder.** We use GCN (Kipf & Welling, 2017), GatedGCN (Bresson & Laurent, 2017), GINE (Hu et al., 2019) and Graph Transformer (Shi et al., 2020) directly, which also server as the patch encoder of Graph MLP-Mixer to see its general uplift effect. Hyperparameter and model configuration are described in Table 7 and Table 8.

**GNN-free patch encoder.** The GNN-free patch encoder reported in Table 2, is an all-MLP architecture. For each graph patches, we embed all node features (bag of nodes), all edge features (bag of edges), then average and readout, based exclusively on MLPs.

$$h_{G_p} = \mathrm{MLP}_3\Big( \sum_{i \in V_p} \mathrm{MLP}_1(h_i) + \sum_{e_{ij} \in E_p} \mathrm{MLP}_2(e_{ij}) \Big) \tag{9}$$

### A.5 DATA AUGMENTATION

Table 12 provides the full results of Table 3.

### A.6 MIXER LAYER

In table 13, GNN-Trans-Encoder is the model whose MLP-Mixer layers are replaced with the same number of standard transformer encoder layers. The rest architecture is the same.

| Model | DA | ZINC | | MolHIV | |
|---|---|---|---|---|---|
| | | MAE | Time(s/epoch) | ROCAUC | Time(s/epoch) |
| GCN-MLP-Mixer | ✗ | 0.1983 ± 0.0026 | 5.6249 | 0.7765 ± 0.0158 | 13.9986 |
| | ✓ | 0.1880 ± 0.0067 | 5.7604 | 0.7865 ± 0.0130 | 14.5451 |
| GatedGCN-MLP-Mixer | ✗ | 0.1625 ± 0.0022 | 5.9534 | 0.7642 ± 0.0188 | 16.5646 |
| | ✓ | 0.1565 ± 0.0023 | 6.1519 | 0.7824 ± 0.0062 | 16.9663 |
| GINE-MLP-Mixer | ✗ | 0.1266 ± 0.0051 | 5.9216 | 0.7569 ± 0.0281 | 14.8949 |
| | ✓ | 0.1171 ± 0.0029 | 5.9922 | 0.7824 ± 0.0131 | 15.5889 |
| GraphTrans-MLP-Mixer | ✗ | 0.1316 ± 0.0043 | 6.3519 | 0.7566 ± 0.0260 | 16.8957 |
| | ✓ | 0.1190 ± 0.0016 | 6.4986 | 0.7805 ± 0.0182 | 17.2023 |

Table 12: Effect of data augmentation (DA): ✗ means no DA and ✓uses DA.

| Model | ZINC | | MolHIV | |
|---|---|---|---|---|
| | Graph MLP-Mixer* | Graph ViT | Graph MLP-Mixer* | Graph ViT |
| GCN- | 0.1352 ± 0.0056 | 0.1743 ± 0.0016 | 0.7849 ± 0.0043 | 0.7580 ± 0.0247 |
| GatedGCN- | 0.1251 ± 0.0080 | 0.1546 ± 0.0024 | 0.7912 ± 0.0133 | 0.7707 ± 0.0067 |
| GINE- | 0.0794 ± 0.0028 | 0.1060 ± 0.0014 | 0.7844 ± 0.0033 | 0.7694 ± 0.0158 |
| GraphTrans- | 0.0831 ± 0.0032 | 0.1160 ± 0.0053 | 0.7823 ± 0.0217 | 0.7750 ± 0.0224 |

Table 13: Mixer Layer vs. Transformer Layer.

## A.7 STATE-OF-THE-ART

For ZINC from Dwivedi et al. (2020), all models have approximately $\sim 500k$ parameters. For MolHIV and MolPCBA, there is no upper limit on the number of parameters. To enable a fair comparison, we set the batch size to 128 for ZINC and MolHIV, and 256 for MolPCBA for all the SOTA GT models and ours, and run all experiments using the same machine.

For SUN (Frasca et al., 2022), due to the huge memory consumption, we use a small batch size of 8 with gradient accumulation. Otherwise, any batch size larger than 8 will lead to out of memory (OOM).

| Model | ZINC | | MolHIV | | | MolPCBA | | |
|---|---|---|---|---|---|---|---|---|
| | MAE ↓ | S/Epoch | ROCAUC ↑ | # Param | S/Epoch | ROCAUC ↑ | # Param | S/Epoch |
| GT | 0.226 ± 0.014 | 9.45 | – | – | – | – | – | – |
| GraphiT | 0.202 ± 0.011 | 8.69 | – | – | – | – | – | – |
| Graphormer | 0.122 ± 0.006 | 105.10 | – | – | – | – | – | – |
| GPS | 0.070 ± 0.004 | 6.30 | 0.7880 ± 0.0101 | 558k | 27.03 | 0.2907 ± 0.0028 | 9,744k | 156.28 |
| SAN | 0.139 ± 0.006 | 17.58 | 0.7775 ± 0.0061 | 714k | 61.94 | 0.2765 ± 0.0042 | 6,062k | 843.02 |
| SAT | 0.094 ± 0.008 | 19.38 | – | – | – | – | – | – |
| GraphTrans | – | – | – | – | – | 0.2761 ± 0.0029 | 4,223k | 360.36 |
| GNN-AK+ | 0.080 ± 0.001 | 8.67 | 0.7961 ± 0.0119 | 414k | 15.20 | 0.2930 ± 0.0044 | 8,693k | 366.04 |
| SUN | 0.084 ± 0.002 | 12.41 | 0.8003 ± 0.0055 | 218k | 32.50 | 0.2616 ± 0.0049 | 14,883k | 1776.96 |
| GNN-MLP-Mixer* (Ours) | 0.0794 ± 0.0028 | 5.88 | 0.7912 ± 0.0133 | 412k | 14.82 | 0.2669 ± 0.0022 | 2,971k | 222.72 |

Table 14: Comparison of our best results from Table 2 with the state-of-the-art Models (missing values from literature are indicated with '-'). For ZINC, all models have approximately $\sim 500k$ parameters.

## A.8 EXPRESSIVITY

We experimentally validate that Graph MLP-Mixer is strictly more powerful than 3-WL by running it on the highly symmetric Circulant Skip Link (CSL) dataset from Murphy et al. (2019) and SR25 (Balcilar et al., 2021), which required GNNs to be strictly more expressive than the 1-WL test and 3-WL respectively to succeed, see Table 16.

| Model | # Params | Peptide-func | | | Peptide-struct | | |
|-------|----------|--------------|--|--|----------------|--|--|
| | | Avg. Precision ↑ | Time(S/Epoch) | Memory(MB) | MAE ↓ | Time(S/Epoch) | Memory(MB) |
| GCN | 508k | 0.5930±0.0023 | – | – | 0.3496±0.0013 | – | – |
| GINE | 476k | 0.5498±0.0079 | – | – | 0.3547±0.0045 | – | – |
| GatedGCN | 509k | 0.5864±0.0077 | – | – | 0.3420±0.0013 | – | – |
| GatedGCN + RWSE | 506k | 0.6069±0.0035 | – | – | 0.3357±0.0006 | – | – |
| Transformer + LapPE | 488k | 0.6326±0.0126 | – | – | 0.2529±0.0016 | – | – |
| SAN + LapPE | 493k | 0.6384±0.0121 | – | – | 0.2683±0.0043 | – | – |
| SAN + RWSE | 500k | 0.6439±0.0075 | – | – | 0.2545±0.0012 | – | – |
| GPS | 504k | 0.6562 ± 0.0115 | 11.83 | – | 0.2515 ± 0.0012 | 11.74 | – |
| GNN-AK+ | 631k | 0.6480 ± 0.0089 | 22.5207 | 7,855 | 0.2736 ± 0.0007 | 22.1169 | 7,634 |
| SUN | 508k | 0.6730 ± 0.0078 | 376.6632 | 18,941 | 0.2498 ± 0.0008 | 384.2698 | 17,215 |
| GCN-MLP-Mixer* | 447k | 0.6736 ± 0.0040 | 18.8352 | 338 | 0.2488 ± 0.0012 | 16.6935 | 332 |
| GatedGCN-MLP-Mixer* | 595k | 0.6790 ± 0.0073 | 19.6521 | 471 | 0.2481 ± 0.0013 | 16.5547 | 478 |
| GINE-MLP-Mixer* | 497k | **0.6846 ± 0.0068** | 19.5324 | 488 | **0.2478 ± 0.0010** | 15.9089 | 482 |
| GraphTrans-MLP-Mixer* | 645k | 0.6814 ± 0.0072 | 18.8322 | 519 | 0.2483 ± 0.0013 | 16.3987 | 513 |

Table 15: Performance on Peptides-func and Peptides-struct (Dwivedi et al., 2022).

| Model | CSL | SR25 |
|-------|-----|------|
| | Accuracy ↑ | Accuracy ↑ |
| GCN-MLP-Mixer* | 1.0000 ± 0.0000 | 1.0000 ± 0.0000 |
| GatedGCN-MLP-Mixer* | 1.0000 ± 0.0000 | 1.0000 ± 0.0000 |
| GINE-MLP-Mixer* | 0.9800 ± 0.0183 | 1.0000 ± 0.0000 |
| GraphTrans-MLP-Mixer* | 1.0000 ± 0.0000 | 1.0000 ± 0.0000 |

Table 16: Results for the CSL (Murphy et al., 2019) dataset and SR25 (Balcilar et al., 2021) dataset.

## A.9 GNN-MLP-Mixer*

Like MLP-Mixer in Computer Vision, GNN-MLP-Mixer is a sequential two-step process. The first step embeds the nodes contained in the graph patches with a MP-GNN and pools the node embedding together to generate a patch representation. The second step combines the patch representations with a mixer layer. However, unlike the original MLP-Mixer, meaningful node representations are difficult to produce due to the high variability of graphs. To improve the expressiveness of GNN-MLP-Mixer, we propose an iterative two-step process by updating alternatively the representation of nodes and patches as follows.

First, the node and edge representations are updated with a MP-GNN applied to each graph patch $G_p(\mathcal{V}_p, \mathcal{E}_p)$ separately and independently:

$$
\begin{aligned}
h_{i,p}^{\ell+1} &= f_h(h_{i,p}^\ell, \{h_{j,p}^\ell | j \in \mathcal{N}(i)\}, e_{ij,p}^\ell) \quad h_{i,p}^{\ell+1}, h_{i,p}^\ell \in \mathbb{R}^d, \\
e_{ij,p}^{\ell+1} &= f_e(h_{i,p}^\ell, h_{i,p}^\ell, e_{ij,p}^\ell) \quad e_{ij,p}^{\ell+1}, e_{ij,p}^\ell \in \mathbb{R}^d,
\end{aligned}
\tag{10}
$$

where $\ell$ is the layer index, $p$ is the patch index, $i, j$ denotes the nodes, $\mathcal{N}(i)$ is the neighborhood of the node $i$ and functions $f_h$ and $f_e$ (with learnable parameters) define the specific MP-GNN architecture.

Second, a fixed-size vector representation of the patch $G_p$ is produced by mean pooling all node vectors in the patch followed by a MLP:

$$
\begin{aligned}
z_p^{l+1} &= \sum_{i \in \mathcal{V}_p} h_i^{l+1} \in \mathbb{R}^d, \\
x_p^{l+1} &= \text{MLP}(z_p^{l+1}) \in \mathbb{R}^d.
\end{aligned}
\tag{11}
$$

Third, the patches represented by $X^{l+1} = \{x_1^{l+1}, ..., x_P^{l+1}\} \in \mathbb{R}^{P \times d}$ are processed with a MLP-Mixer layer:

$$
\begin{aligned}
U^{l+1} &= X^{l+1} + (W_2^{l+1} \sigma(W_1^{l+1} \text{LayerNorm}(A_\mathcal{D}^P X^{l+1}))) \in \mathbb{R}^{P \times d} \quad &\text{Token mixer,} \\
Y^{l+1} &= U^{l+1} + (W_4^{l+1} \sigma(W_3^{l+1} \text{LayerNorm}(U^{l+1})^T))^T \in \mathbb{R}^{P \times d} \quad &\text{Channel mixer,}
\end{aligned}
\tag{12}
$$

Finally, the node representations are updated using both the outputs of the MP-GNN layer and the MLP-Mixer layer:

$$
\begin{aligned}
\hat{h}_{i,p}^{l+1} &= M^{l+1}h_{i,p}^{l+1} + N^{l+1}Y_p^{l+1} \in \mathbb{R}^d, \\
\hat{h}_i^{l+1} &= \underset{\{p \mid i \in G_p\}}{\mathrm{Mean}}\ \hat{h}_{i,p}^{l+1} \in \mathbb{R}^d, \\
h_{i,p}^{l+1} &\leftarrow \hat{h}_i^{l+1} \in \mathbb{R}^d,
\end{aligned}
\tag{13}
$$

These node embeddings serve as the input of the next layer and the iterative process goes back to the first step above.

## A.10 COMPLEXITY ANALYSIS

For each graph $G = (\mathcal{V}, \mathcal{E})$, with $N = |\mathcal{V}|$ being the number of nodes and $E = |\mathcal{E}|$ being the number of edges, the METIS patch extraction takes $O(E)$ runtime complexity, and outputs graph patches $\{G_1, ..., G_P\}$, with $P$ being the pre-defined number of patches. Accordingly, we denote each graph patch as $G_p = (\mathcal{V}_p, \mathcal{E}_p)$, with $N_p = |\mathcal{V}_p|$ being the number of nodes and $E_p = |\mathcal{E}_p|$ being the number of edges in $G_p$. After our one-hop overlapping adjustment, the total number of nodes and edges of all the patches are $(N_U = \sum_p N_p) \le 2N$ and $(E_U = \sum_p E_p) \le 2E$, respectively. Assuming base GNN has $O(E)$ runtime complexity, our patch embedding module has $O(E_U)$ runtime complexity. For the Mixer Layer, the complexity is $O(P)$.

## A.11 OVER-SQUASHING

We illustrate experimentally the over-fitting property that can be produced by over-squashing with the synthetic TreeNeighbour dataset from (Alon & Yahav, 2020) and a real-world long-range dataset borrowed from (Dwivedi et al., 2022) in Figure 6 and Table 17.

For the synthetic TreeNeighbour dataset, we run the experiments with GCN, GGCN and Graph MLP-Mixer with the number of layers being the double size as the tree depth and a hidden size of 128, and it can be observed that there is no issue for the GNNs to overfit this dataset. In (Alon & Yahav, 2020), the authors use a number of layers equal to the tree depth+1 and a small hidden size of 32, which does not provide enough learning capacity to the networks and thus under-fits the dataset.

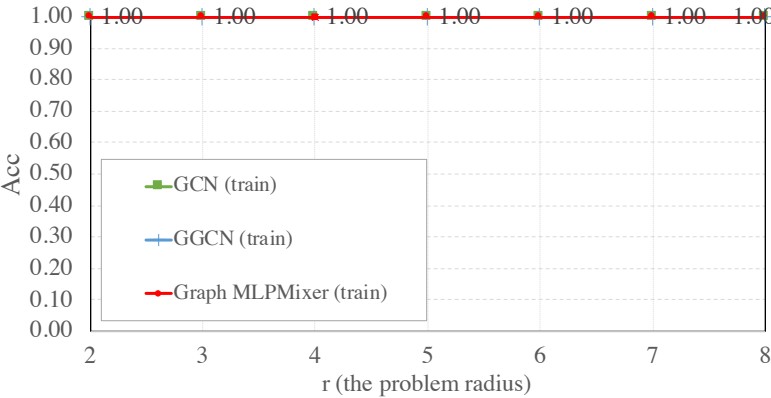

Figure 6: Train Accuracy across problem radius (tree depth) in the NEIGHBORSMATCH problem (Alon & Yahav, 2020).

We confirmed this observation with the real-world long-range graph dataset, peptide-func taken from (Dwivedi et al., 2022) with a mean value of 57 for the graph diameter of these large graphs. The same result occurs, the GCN, GGCN and Graph MLP-Mixer are able to over-fit the dataset at almost 100%.

Figure 7 presents the generalization performance of GNNs on the TreeNeighbour dataset specifically designed to analyze synthetically the property of datasets with long-range dependencies.

| Model | Training Accurary |
|---|---|
| GGCN | 0.9991 |
| GCN | 0.9993 |
| Graph MLP-Mixer | 0.9993 |

Table 17: Training Accuracy in the Peptides-func problem (Dwivedi et al., 2022).

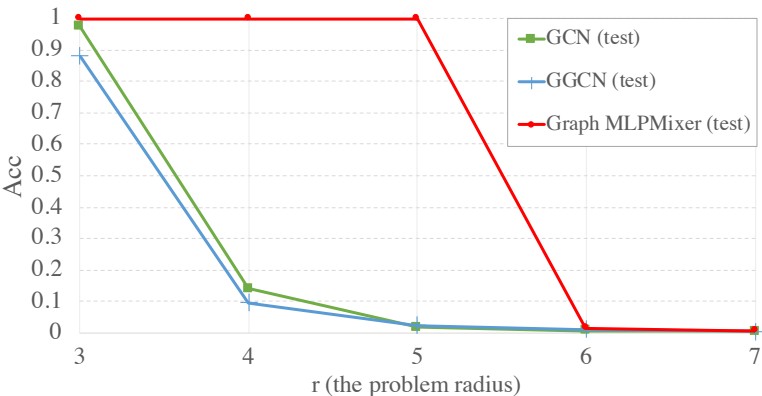

Figure 7: Test Accuracy across problem radius (tree depth) in the NEIGHBORSMATCH problem (Alon & Yahav, 2020).

