# OpenReview forum: "Graph MLP-Mixer"
_ICLR.cc/2023/Conference — Submitted to ICLR 2023_

### Official Review · Reviewer_jedP · 2022-10-15

**Confidence:** 4
**Correctness:** 2
**Technical Novelty And Significance:** 3
**Empirical Novelty And Significance:** 2
**Recommendation:** 6

**Clarity, Quality, Novelty And Reproducibility:**

\
**Clarity**

While the paper seems well-organised, its clarity can be improved in technical reasoning aspects.

Specifically, more clarity on the relationship between oversquashing and overfitting is needed.

The relationship should also be positioned with existing literature which has argued that oversquashing actually leads to underfitting [2].

[2] On the Bottleneck of Graph Neural Networks and its Practical Implications, In ICLR'21.

\
**Quality**

The paper's quality can be improved through more empirical evaluations in support of the claims.

Experiments on oversquashing are needed to support tha claim that Graph MLP-Mixer can mitigate oversquashing, e.g., TreeNeighboursMatch problem in prior work [2] (Section 4.1 in the paper).

More experiments are needed to confirm that Graph MLP-Mixer can capture long-range dependendcies, e.g., the authors could consider Code2 dataset used in prior work [3] (Section 5.4 in the paper).

[3] Representing Long-Range Context for Graph Neural Networks with Global Attention, In NeurIPS'21.

\
**Novelty**

This is the first adaptation of MLP-Mixer to graph structured data.

MLP-Mixer can potentially address limitations of popular models (GNNs and GTs) on graphs.

\
**Reproducibility**

The main part and the supplementary part include enough material, e.g., dataset details, baselines with references, hyperparameters, effect of different components, for an expert to replicate the results of the paper.

___

**Strength And Weaknesses:**

\
**Strengths**

\+ The paper is well-organised and Table 1 clearly lists key differences of MLP-mixer for images and the proposed adaptation for graphs.

\+ The proposed method can potentially address important limitations of both GNNs and GTs.


\
**Weaknesses**

\- The authors claim that GNNs suffering from oversquashing leads to overfitting while existing literature [2] has shown that it actually leads to underfitting.

[2] On the Bottleneck of Graph Neural Networks and its Practical Implications, In ICLR'21.

\- The authors attribute the poor long-range modelling ability of very deep GNNs to oversquashing but the poor performance could be due to a combined effect of multiple factors (e.g., oversquashing, oversmoothing, vanishing gradients, etc.).

\- There are no experiments to support the claim that Graph MLP-Mixer can capture long-range interactions in graph datasets.

___

**Summary Of The Paper:**

While graph neural networks (GNNs) and graph transformers (GTs) have been popular for graph representation learning recently, they have their own limitations:
* GNNs, especially those based on message passing, are poor at modelling long-range dependencies and
* GTs suffer from quadratic computation complexity.

This paper adapts a specialised multi-layer perceptron (MLP) architecture, viz., MLP-Mixer, initially proposed in the image domain [1], to graph-structured data for modelling long-range interactions with linear computational cost.

[1] MLP-Mixer: An all-MLP Architecture for Vision, In NeurIPS'21.

Experiments on benchmark graph datasets demonstrate the effectiveness of the proposed method.

___

**Summary Of The Review:**

While the proposed method is potentially promising, more clarity and empirical evaluation are needed in support of the claims made in the paper.

___


Update:
I have read all the reviews and their responses. I have raised my score from 5 to 6.

---

> ### Author Response · Authors · 2022-11-15
> **Response to Reviewer jedP (Part 1/2)**
>
> We thank the reviewer for her/his careful reading and comments regarding our work. Please, see below our answer to the raised comments/questions and ---find the revisions in blue color in the rebuttal manuscript.
>
> ---
>
> > **Reviewer**: The authors claim that GNNs suffering from over squashing leads to overfitting while existing literature [1] has shown that it actually leads to underfitting.
> > While the paper seems well-organized, its clarity can be improved in technical reasoning aspects. Specifically, more clarity on the relationship between oversquashing and overfitting is needed. The relationship should also be positioned with existing literature which has argued that oversquashing actually leads to underfitting [1].
> > Experiments on over squashing are needed to support the claim that Graph MLP-Mixer can mitigate over squashing, e.g., TreeNeighboursMatch problem in prior work [1] (Section 4.1 in the paper).
>
> **Authors**: The over-squashing issue has been a long-standing limitation for Recurrent Neural Networks (RNNs) and GNNs. This issue limits RNNs to successfully perform e.g. the translation task on sequences longer than 20 words (Figure 2 in [2]), although the reception field only grows linearly at each layer. For GNNs, the over-squashing issue is worse than RNNs as the reception field can grow exponentially (e.g. trees). But this unfavorable property does not imply that RNNs/GNNs always under-fit. We agree that GNNs under-fit when the learning capacity of the networks is limited, i.e. not enough layers and small hidden dimensions. However, for real-world experiments, it is standard practice to select a large number of layers and large hidden dimensions to achieve good performance [3]. In this context, over-squashing leads to the over-fitting problem.
>
> We can demonstrate experimentally this over-fitting property with the synthetic TreeNeighbour dataset from [1] mentioned by the Reviewer, and with a real-world long-range dataset borrowed from [4]. Results can be found in Annex A.10 (Figure 6 and Table 17). For the synthetic TreeNeighbour dataset, we ran the experiments with GCN, GGCN and Graph MLP-Mixer with a number of layers being the double size as the tree depth and a hidden size of 128, and it can be observed that the GNNs have no difficulty to over-fit this dataset. In [1], the authors use a number of layers equal to the tree depth+1 and a small hidden size of 32, which does not provide enough learning capacity to the networks and thus under-fits the dataset. We confirmed this observation with the real-world long-range graph dataset, Peptides-func taken from [4] with a mean value of 57 for the graph diameter of these large graphs. The same result occurs; the GCN, GGCN and Graph MLP-Mixer are able to over-fit the dataset at almost 100%.
>
> In summary, under-fitting/over-fitting depend more on the learning capacity of the network than the over-squashing issue. Actually, the important question is less about under/over-fitting but more on the generalization of GNNs for datasets with long-range dependencies s.a. the TreeNeighbour dataset which has been designed to analyze synthetically this question. Figure 7 shows that standard message-passing (MP) GNNs like GCN and GGCN are not able to generalize well because of over-squashing that squeezes too much information coming from the exponential growth of the tree reception field. Compared to MP-GNNs, Graph MLP-Mixer is able to mitigate the problem of over-squashing by transmitting the long-distance information directly with the mixer layer, instead of applying multiple times of MP and incurring an exponentially growing receptive field.

---

> > ### Author Response · Authors · 2022-11-15
> > **Response to Reviewer jedP (Part 2/2)**
> >
> > ---
> >
> > > **Reviewer**: The authors attribute the poor long-range modeling ability of very deep GNNs to over squashing but the poor performance could be due to a combined effect of multiple factors (e.g., oversquashing, oversmoothing, vanishing gradients, etc.).
> >
> > **Authors**: We fully agree with the reviewer that the poor long-range modeling ability of deep GNNs can be caused by the combined effect of multiple factors such as vanishing gradients, poor isomorphism expressivity, over-squashing, etc. In this work, we wanted to focus our effort on a recent trend in GNNs, for which GPS [4] represents the most recent publication at NeurIPS’22. In this trend, GNN architectures aim at alleviating over-squashing by combining standard local aggregation operation (traditional message-passing GNNs like GCNs) and global attention mechanism (Transformer-based architectures) to achieve SOTA performances, particularly on large graph datasets which require long-distance modeling [4]. But global attention is expensive because of the computation of matching scores (quadratic complexity), making Graph Transformers intractable for large graphs. [5] suggests to use Performer [6] to reduce the complexity to linear, but this introduces an approximation of the attention mechanism.
> >
> > We propose an alternative to this line of work based on the MLP-Mixer architecture that is naturally designed to capture long-distance interactions (all patches directly communicate with each other) and reduce the complexity to linear (w.r.t. the number of patches) without any approximation. By design, the Graph MLP-Mixer architecture can decrease the over-squashing issue in GNNs while being able to capture local graph topology. Additional experiments on the synthetic SR25 dataset (see Appendix A.8) also demonstrates that the proposed architecture is highly expressive to distinguish at least 3-WL non-isomorphic graphs, while keeping a low linear complexity (GNN-AK+ [7], Table 1, requires the use of PPGN [8] to solve SR25 which requires O(N^2) for memory and O(N^3) for speed, with N being the number of nodes).
> >
> > ---
> >
> > > **Reviewer**: There are no experiments to support the claim that Graph MLP-Mixer can capture long-range interactions in graph datasets.
> >
> > **Authors**: We have provided additional experiments on Long Range Graph Benchmark (LRGB) [4] to verify that Graph MLP-Mixer is able to capture long-range interactions. Peptides-func and Peptides-struct are two graph-level prediction datasets, consisting of 15,535 graphs with a total of 2.3 million nodes. They are better suited to evaluate models enabled with long-range dependencies, as they contain larger graphs and more data points.
> >
> > The performance is reported in Table 2 and Table 5 in the rebuttal revision version and in Table 15 in the Appendix. We summarize the main results below.
> >
> > (1) Graph MLP-Mixer achieves SOTA with the best scores of 0.6846 on Peptides-func and 0.2478 on Peptides-struct (Table 5), demonstrating the ability of the model to best capture long-range relationships.
> >
> > (2) Compared with MP-GNNs (Table 2), Graph MLP-Mixer significantly outperforms the base MP-GNNs; we can observe an average 0.030 Average Precision improvement on Peptides-func and an average 0.030 MAE decrease on Peptides-struct. Compared with SOTA models, Graph MLP-Mixer also ranks among the best (Table 5).
> >
> > (3) Graph MLP-Mixer offers a low-cost architecture to the related architectures, GNN-AK+ [7] and SUN [9]. For example, SUN [9] gives similar performance to Graph MLP-Mixer, 0.6730 on Peptides-func and 0.2498 on Peptides-struct, but requires 30x memory and 18x training time (Table 15). Graph MLP-Mixer is cost-efficient and scalable with large graphs.
> >
> > ---
> >
> > > **Reviewer**: The paper's quality can be improved through more empirical evaluations in support of the claims.
> >
> > **Authors**: We hope we have addressed most of the questions. Please, let us know if you have additional questions.
> >
> > ---
> >
> > References:
> >
> > [1] Alon, Yahav, On the Bottleneck of Graph Neural Networks and its Practical Implications, ICLR 2021.
> >
> > [2] Bahdanau, Cho, Bengio, Neural machine translation by jointly learning to align and translate, 2014
> >
> > [3] Belkin et al, Reconciling modern machine-learning practice and the classical bias–variance trade-off, PNAS 2019
> >
> > [4] Dwivedi et al, Long range graph benchmark, NeurIPS 2022
> >
> > [5] Rampášek et al, Recipe for a general, powerful, scalable graph transformer, NeurIPS 2022
> >
> > [6] Choromanski et al, Rethinking attention with performers, ICLR 2021
> >
> > [7] Zhao, Jin, Akoglu, Shah, From Stars to Subgraphs: Uplifting Any GNN with Local Structure Awareness, ICLR 2022
> >
> > [8] Maron, Ben-Hamu, Serviansky, Lipman, Provably powerful graph networks, NeurIPS 2019
> >
> > [9] Understanding and Extending Subgraph GNNs by Rethinking Their Symmetries, Frasca, Bevilacqua, Bronstein, Maron, NeurIPS 2022

---

> > > ### Comment · Reviewer_jedP · 2022-11-18
> > > **Thanks for the Response**
> > >
> > > Dear Authors,
> > >
> > > Thanks for the response and the experiments on additional datasets (e.g., LRGB).
> > >
> > > I have read all the reviews and their responses. I have raised my rating from 5 to 6. The paper can still be improved in terms of quality.
> > >
> > > Suggestions for further improvement:
> > > * It is important to study the effect of positional information (Figure 3) on the new datasets (Peptides-func and Peptides-struct). Figure 3 gives a hint that positional information can have marginal benefits for some datasets (e.g., MolHIV).
> > > * Since the new datasets require long-range dependency modelling with 50+ graph diameters, it is also important to study the effects of different hop sizes (Figure 2 and Table 10) on Peptides-func and Peptides-struct. What hop sizes give the best results on the new datasets? Is there potential for further improvement, e.g., by increasing the hop size?

---

> > > > ### Author Response · Authors · 2022-11-18
> > > > **Thank you & New Experiments**
> > > >
> > > > We thank the reviewer very much for her/his response and for the new suggestions to improve the paper’s quality.
> > > >
> > > > ---
> > > > >**Reviewer**: It is important to study the effect of positional information (Figure 3) on the new datasets (Peptides-func and Peptides-struct). Figure 3 gives a hint that positional information can have marginal benefits for some datasets (e.g., MolHIV). Since the new datasets require long-range dependency modelling with 50+ graph diameters, it is also important to study the effects of different hop sizes (Figure 2 and Table 10) on Peptides-func and Peptides-struct. What hop sizes give the best results on the new datasets? Is there potential for further improvement, e.g., by increasing the hop size?
> > > >
> > > > **Authors**: We ran the experiments suggested by the Reviewer on the effects of positional information and the k-hop size extension. See updated Figure 3 and Figure 2 in the new version of the manuscript.
> > > >
> > > > **Positional encoding (PE)**: Let us first remember that it was proved in [10,11] that unique and equivariant PE increases the representation power of any MP-GNN (PE leads to GNNs strictly more powerful than the 1-WL test). PE is thus important from a theoretical point of view but, unfortunately, theory does not provide any guidance on the choice of PE for a given graph dataset and task. Consequently, the choice of PE is today arbitrary and is selected by trial-and-error experiments such as the 2022 NeurIPS paper [5] and 2022 ICML work [12] to cite the most recent PE-based GNNs. In our work, we designed the Graph MLP-Mixer model with arbitrary node PE and patch PE in order to add back positional information that is naturally lost when working with graphs (graphs have no canonical positioning of vertices). Then, for ZINC, MolHIV and Peptides-func/struct, we used the k-step Random Walk for the node PE and the patch PE (with different adjacency matrices for the nodes and the patches). This results into a significant improvement for ZINC (e.g. MAE from 0.461 to 0.119) and moderate increases for MolHIV (e.g. Acc from 0.763 to 0.780, +1.7%) and Peptides-func (e.g. Acc from 0.667 to 0.684, +1.7%). As a final word, PE is certainly useful to improve the quality of GNN prediction given the theory and the increased number of published works on this topic, but more (mathematical) progress is needed to identify better choices and provide consistent result improvement.
> > > >
> > > > **K-hop extension**: Without any extension of the patch support, it is not possible to capture all edges from the original graph as we lose the edges that have been cut during the graph partitioning process with METIS. This is why the performance is the worst without support extension (0-hop) vs. k-hop for k>=1. This is true for all the three datasets ZINC, MolHIC and Peptides-func and for all GNN-MLP-Mixer models (except GCN for Peptides-func). With overlapping patches (k-hop, k>=1), the best performance is obtained at a sweet spot that naturally depends on the dataset at hand, and needs to be evaluated as any hyper-parameter. However, in practice, we used the value of k=1 to keep the computational complexity low and without much performance drop.
> > > >
> > > > ---
> > > >
> > > > Thanks again for your suggestion. Please, let us know if you have more recommendations.
> > > >
> > > >
> > > > Additional References:
> > > >
> > > > [10] Murphy et al, Relational pooling for graph representations, ICML 2019
> > > >
> > > > [11] Loukas, What graph neural networks cannot learn: depth vs width, ICLR 2020
> > > >
> > > > [12] Lim et-al, Sign and basis invariant networks for spectral graph representation learning, ICLR 2022

---

### Official Review · Reviewer_Fxzu · 2022-10-24

**Confidence:** 3
**Clarity, Quality, Novelty And Reproducibility:** The paper is well written, and the me…
**Correctness:** 3
**Technical Novelty And Significance:** 3
**Empirical Novelty And Significance:** 3
**Recommendation:** 5

**Strength And Weaknesses:**

### Strengths
1. The paper proposes to apply MLP mixer to graph learning.
2. The authors introduce a method for extracting patches from the graph and adding positional information for the patch.
3. The experimental results show that the graph MLP-mixer can improve the performance of the existing method.

### Weaknesses
1. The paper claims that the method can get rid of the over-squashing problem of the GNN model. It would be great to see the larger scale dataset with longer-range connections on the OGB dataset, such as OGBG-code2.
2. The paper selects METIS as the graph partitioning algorithm to generate patches. It can be interesting to have the ablation study for how many benefits the METIS can provide against random partitioning.
3. The Molpcba results in Table 5 do not seem very convincing, it would be better to see a fair comparison by having a similar number of parameters.



**Summary Of The Paper:**

The paper explores the MLP Mixer based approach on graph learning. It claims that the architecture can get rid of over-squashing or high complexity problem for long-range connections.

**Summary Of The Review:**

It is an interesting exploration of the MLP mixer for graph learning and outperforms multiple existing works, but the experiments need to be improved.

---

> ### Author Response · Authors · 2022-11-15
> **Reponse to Reviewer Fxzu (Part 1/2)**
>
> We thank the reviewer for her/his careful reading and comments regarding our work. Please, see below our answer to the raised comments/questions and find the revisions in blue color in the rebuttal manuscript.
>
> ---
>
> > **Reviewer**: The paper claims that the method can get rid of the over-squashing problem of the GNN model. It would be great to see the larger scale dataset with longer-range connections on the OGB dataset, such as OGBG-code2.
>
> **Authors**: We have provided additional experiments on Long Range Graph Benchmark (LRGB) [1] to verify that Graph MLP-Mixer is able to capture long-range interactions. Peptides-func and Peptides-struct are two graph-level prediction datasets, consisting of 15,535 graphs with a total of 2.3 million nodes. They are better suited to evaluate models enabled with long-range dependencies, as they contain larger graphs and more data points.
>
> The performance is reported in Table 2 and Table 5 in the rebuttal revision version and in Table 15 in the Appendix. We summarize the main results below.
>
> (1) Graph MLP-Mixer achieves SOTA with the best scores of 0.6846 on Peptides-func and 0.2478 on Peptides-struct (Table 5), demonstrating the ability of the model to best capture long-range relationships.
>
> (2) Compared with MP-GNNs (Table 2), Graph MLP-Mixer significantly outperforms the base MP-GNNs; we can observe an average 0.030 Average Precision improvement on Peptides-func and an average 0.030 MAE decrease on Peptides-struct. Compared with SOTA models, Graph MLP-Mixer also ranks among the best (Table 5).
>
> (3) Graph MLP-Mixer offers a low-cost architecture to the related architectures, GNN-AK+ [2] and SUN [3]. For example, SUN [3] gives similar performance to Graph MLP-Mixer, 0.6730 on Peptides-func and 0.2498 on Peptides-struct, but requires 30x memory and 18x training time (Table 15). Graph MLP-Mixer is cost-efficient and scalable with large graphs.
>
> ---
>
> > **Reviewer**: The paper selects METIS as the graph partitioning algorithm to generate patches. It can be interesting to have the ablation study for how many benefits the METIS can provide against random partitioning.
>
> **Authors**: We ran the suggested experiment by the reviewer, that is to quantify how many benefits the METIS can provide against random partitioning. We reported the results in Table 11 in the Appendix. For random graph partition, nodes are randomly assigned to a pre-defined number of patches. For both METIS and random partition, we re-generated graph patches at each epoch. We showed that using METIS as the graph partition algorithm consistently gives better performance than random node partition, which corresponds to our intuition that nodes and edges composing a patch should share similar semantic or information. For example, the best performance on the ZINC datasets is 0.0794 with METIS partition and 0.1057 with random graph partition. Similarly, the best performance on MolHIV is 0.7912 and 0.7791 with METIS and random partition, respectively. Nevertheless, it is interesting to see that random graph partitioning is still able to achieve reasonable results, which shows that the performance of the model is not solely supported by the quality of the patches.
>
> ---

---

> > ### Author Response · Authors · 2022-11-15
> > **Response to Reviewer Fxzu (Part 2/2)**
> >
> >
> > > **Reviewer**: The Molpcba results in Table 5 do not seem very convincing, it would be better to see a fair comparison by having a similar number of parameters.
> >
> > **Authors**: We agree with the reviewer that our model does not perform on MolPCBA as well as the other models such as GPS [4] or GNN-AK+ [2]. We believe that MolPCBA is a little more convoluted dataset than the other ZINC, molHIV, Peptides-func and Peptides-struct datasets. MolPCBA aims at predicting 128 binary classification tasks simultaneously while ZINC, molHIV, Peptides-func and Peptides-struct targets one single classification or regression task. In other words, this multiple-objective task requires more architectural grid-search to succeed on average. For example, GNN-AK+ architecture is designed with three central encodings; a centroid encoding, a subgraph encoding, and a context encoding (Figure 1 [2]). Neither the centroid encoding nor the context encoding are used in the GitHub code [5] and other model hyper-parameters are grid-searched to optimize the results. Because of the lack of computational resources, we did not perform any grid-search optimization and most of our hyper-parameters are similar across the experiments.
> >
> > In addition, we would like to bring to the attention of the reviewer that we have recently improved our model to be competitive to SOTA models in most datasets, except for MolPCBA. We enhanced the original network by alternatingly applying Patch Encoder and Mixer Layer and we called this updated model Graph MLP-Mixer*. The original Graph MLP-Mixer was sequential; the Patch Encoder was first applied to embed graph patches into a vectorial representation (no further update was possible) and then the Mixer Layer combined the patch representations. The updated version improves the node and the patch representations alternatingly, which results in better performance as more information is exchanged between the nodes and the patches. Other components of the proposed architecture remain the same. A detailed description of the improved model can be found in Appendix A.9 and the results are reported  in Table 5, Table 14 and 15. Specifically, for small molecular graphs, Graph MLP-Mixer* achieved 0.0794 on ZINC, and 0.7912 on MolHIV, achieving comparable performance of SOTA models (ZINC: 0.070 (GPS) 0.084 (SUN); MolHIV: 0.7880 (GPS), 0.8003 (SUN)). For larger molecular graphs, Graph MLP-Mixer* showed superiority compared to both MP-GNNs and SOTA models on Peptides-func and Peptides-struct as discussed above.
> >
> > ---
> >
> > Please, let us know if you have additional questions.
> >
> > References:
> >
> > [1] Dwivedi et al, Long range graph benchmark, NeurIPS 2022
> >
> > [2] Zhao et al, From stars to subgraphs: Uplifting any GNN with local structure awareness, ICLR 2022
> >
> > [3] Understanding and Extending Subgraph GNNs by Rethinking Their Symmetries, Frasca, Bevilacqua, Bronstein, Maron, NeurIPS 2022
> >
> > [4] Rampášek et al, Recipe for a general, powerful, scalable graph transformer, NeurIPS 2022
> >
> > [5] [https://github.com/LingxiaoShawn/GNNAsKernel/blob/main/train/configs/molpcba.yaml](https://github.com/LingxiaoShawn/GNNAsKernel/blob/main/train/configs/molpcba.yaml)**

---

> > > ### Author Response · Authors · 2022-11-19
> > > **Follow-up on Rebuttal Feedback**
> > >
> > > Dear Reviewer Fxzu,
> > > &nbsp;
> > >
> > > We thank you again for taking your time reviewing our work. We put our best effort to answer your questions.
> > > &nbsp;
> > >
> > > We would appreciate to know if you need further clarification or have additional question.
> > > &nbsp;
> > >
> > > If you are satisfied with our answers, please consider revising your score.
> > > &nbsp;
> > >
> > > Best regards,
> > >
> > > The Authors

---

### Official Review · Reviewer_oFvZ · 2022-10-26

**Confidence:** 3
**Clarity, Quality, Novelty And Reproducibility:** I wrote a point about novelty at the …
**Correctness:** 2
**Technical Novelty And Significance:** 2
**Empirical Novelty And Significance:** 2
**Recommendation:** 5

**Strength And Weaknesses:**

pros:

[+] The paper is well-written and organized.  Related works and the background concepts are described enough to understand the main idea of Graph MLP-Mixer.

[+] The problem definition and the development process are quite logical and straightforward.


---

cons:

[-] It would be nice if there were more analyzes or novel modules for graphs.
It seems like a simple extension of MLP-Mixer with a little bit classic graph clustering module.



**Summary Of The Paper:**

Message passing-based GNNs suffer from over-squashing and poor long-range dependencies problems.

Graph Transformers have been proposed because the global attention mechanism can alleviate the above problems.

However, Graph Transformers have a computational complexity proportional to the square of the number of nodes, just like transformers in other fields.

In this paper, the authors propose an MLP-Mixer model for graph-structured data to overcome these issues.

The original MLP-Mixer, which is also unaffected by over-squashing and weak long-range dependencies, takes Multi-layer perceptrons (MLPs) in the place of the attention module.

Contrary to visual data, which has a grid structure, graphs have an irregular structure, so they use a clustering algorithm to extract the patches for MLP-Mixer.

**Summary Of The Review:**

If the authors provide a clear novel point or analysis on this, I'll raise the score.

---

> ### Author Response · Authors · 2022-11-15
> **Response to Reviewer oFvZ (Part 1/2)**
>
> We thank the reviewer for her/his careful reading and comments regarding our work. Please, see below our answer to the raised comments/questions and find the revisions in blue color in the rebuttal manuscript.
>
> ---
>
> > **Reviewer**: It would be nice if there were more analyzes or novel modules for graphs. It seems like a simple extension of MLP-Mixer with a little bit classic graph clustering module.
>
> **Authors**:
>
> **Novelty**: The main novelty of this work is to design a GNN architecture that is not limited by poor long-range dependencies while keeping a low computational complexity. Standard MP-GNNs have a representation limitation in terms of isomorphism expressivity that can be improved with WL-GNNs and graph positional encoding. However, none of these improved networks are able to deal well with long-distance communication between nodes. Communication between far away nodes is usually achieved by stacking multiple layers, for which the support of message-passing functions increases by one-hop at each layer. But increasing the number of layers implies an exponential growth of the receptive field of the nodes, which results in squeezing a large set of neighbor-vectors into one single vector. Such an issue, known as over-squashing, is today the major limitation of GNN design. To address this problem, Graph Transformers leverage the full-attention mechanism but this affects badly the computational efficiency with a quadratic complexity O(N^2), N being the #nodes [1,2]. The main novelty of this paper is to leverage the recent MLP-Mixer architecture from computer vision that solves the issue of long-distance dependencies with low-cost operations.
>
> **Contribution**: The main contribution is to generalize ViT/MLP-Mixer to graphs in order to design novel GNNs that are simultaneously 1) expressive in terms of graph isomorphism, 2) capture long-range dependencies, and 3) offer memory and speed efficiency. To achieve these properties, our method addresses a few main generalization challenges such as the lack of canonical position in graphs. Addressing these challenges opens the possibility of applying more patch-based approaches to graph data, which can lead to subsequent works which also efficiently capture long range dependencies.
>
> **Numerical experiments**: The proposed Graph MLP-Mixer* demonstrates competitive experimental results for small graphs (ZINC, MolHIV) [3,4] and SOTA performances for large graphs (Peptide-func, Peptide-struct) [5]. More precisely, (as shown in Table 5) Graph MLP-Mixer achieved 0.0794 on ZINC, and 0.7912 on MolHIV, achieving comparable performance of SOTA models (ZINC: 0.070 (GPS) 0.084 (SUN); MolHIV: 0.7880 (GPS), 0.8003 (SUN)). Graph MLP-Mixer* showed best performances with 0.6846 on Peptides-func and 0.2478 on Peptides-struct compared to 0.6730 and 0.2498 achieved with SUN, the second best model. Besides, Graph MLP-Mixer* offers better complexity and scalability than related models. For e.g. MolPCBA, Graph Transformer (GT) models require 843sec/epoch and 360sec/epoch respectively vs. 222sec/epoch for Graph MLP-Mixer (Table 14). And for larger graphs, related GNNs such as the recent 2022 ICLR GNN-AK+ [6] and the 2022 NeurIPS SUN [7] require 15x and 30x memory respectively compared to Graph MLP-Mixer (Table 15). Finally, we demonstrated the isomorphism expressivity of Graph MLP-Mixer on the synthetic SR25 dataset (see Appendix A.8), which requires GNNs with at least 3-WL discriminative power to distinguish non-isomorphic graphs. Graph MLP-Mixer was successful, independently of the GNN encoder and keeping a low linear complexity (GNN-AK+ [6], Table 1, requires the use of PPGN [8] to solve SR25 which requires O(N^2) for memory and O(N^3) for speed), N being the number of nodes.

---

> > ### Author Response · Authors · 2022-11-15
> > **Response to Reviewer oFvZ (Part 2/2)**
> >
> > **Analysis**: Generalizing GNNs to ViT/MLP–Mixer from images to graphs is not straightforward as we detailed the challenges of this extension in Section 2.
> >
> > 1) Patch Definition/Extraction: The graph patch extraction part requires a fast and accurate graph partitioning algorithm that is able to identify meaningful graph sub-structures. METIS provides a good trade-off speed and accuracy in terms of clustering quality. We demonstrated the quality and impact of METIS in Table 11 against random partitioning. However, METIS is not able to provide different clusters at each epoch during training, which may lead to over-fitting. To alleviate this issue, we designed a fast dropping edge technique that produces distinct clusters at each epoch. Besides, unlike non-overlapping image patches, we propose to overlap graph patches to preserve all the edge information. We validate the effectiveness of METIS patch extraction (Table 11), fast dropping edge (Table 3), and patch overlapping (Figure 2). Overall, the patch extraction module can be regarded as a graph tokenizer, which is applicable to any token/patch-based architectures beyond MLP-Mixer, to design more expressive GNNs. It is a critical component to align graph representation learning with CV/NLP under a unified framework
> > 2) Patch Encoder: Unlike the standard patch encoder in ViT/MLP–Mixer in computer vision, the graph patch encoder is updated not only with the node features but also with the patch representation of the mixer layer, which provides long-distance information to best compute the representation of nodes.
> > 3) Positional Information: Another generalization challenge is the graph positional encoding (PE) for the nodes and the patches. Unlike images, which are implicitly supported with an Euclidean grid, graphs have no canonical node ordering. The choice of the PEs is thus critical to get good performances as we discussed in Section 1. We therefore design two explicit positional encoding (PE): Node PE and Patch PE. Figure 3 showed the impact of PE is the architecture design.
> > 4) MLP-Mixer: We replace the Mixer layer in Graph MLP-Mixer with the standard Transformer layer as in ViT, keeping the rest the same (denoted as Graph ViT). Interestingly, experiments also showed that Graph MLP-Mixer architecture performs better than Graph ViT, see Table 13. It seems that the sparse attention mechanism is more difficult to train in the context of graphs, as observed in [9].
> >
> > **Conclusion**: We hope we have clarified the novelty and impact of the proposed GNN model in the context and development of actual GNNs. Particularly, we believe that the proposed model is the first architecture that properly solves the over-squashing issue while keeping a low computational complexity and being highly expressive. Please, let us know if you have additional questions.
> >
> > References:
> >
> > [1] Kreuzer, Beaini, Hamilton, Letourneau, Tossou, Rethinking Graph Transformers with Spectral Attention, NeurIPS 2021
> >
> > [2] Ying, Cai, Luo, Zheng, Ke, He, Shen, Liu, Do Transformers Really Perform Bad for Graph Representation?, NeurIPS 2021
> >
> > [3] Dwivedi et al, Benchmarking graph neural networks, 2020
> >
> > [4] Hu et al, Open graph benchmark: Datasets for machine learning on graphs, 2020
> >
> > [5] Dwivedi et al, Long range graph benchmark, NeurIPS 2022
> >
> > [6] Zhao, Jin, Akoglu, Shah, From Stars to Subgraphs: Uplifting Any GNN with Local Structure Awareness, ICLR 2022
> >
> > [7] Understanding and Extending Subgraph GNNs by Rethinking Their Symmetries, Frasca, Bevilacqua, Bronstein, Maron, NeurIPS 2022
> >
> > [8] Maron, Ben-Hamu, Serviansky, Lipman, Provably powerful graph networks, NeurIPS 2019
> >
> > [9] Dwivedi, Bresson, A generalization of transformer networks to graphs, 2020

---

> > > ### Author Response · Authors · 2022-11-19
> > > **Follow-up on Rebuttal Feedback**
> > >
> > > Dear Reviewer oFvZ,
> > > &nbsp;
> > >
> > > We thank you again for taking your time reviewing our work. We put our best effort to answer your questions.
> > > &nbsp;
> > >
> > >
> > > We would appreciate to know if you need further clarification or have additional question.&nbsp;
> > > &nbsp;
> > >
> > > If you are satisfied with our answers, please consider revising your score.
> > > &nbsp;
> > >
> > >
> > > Best regards,
> > >
> > > The Authors

---

### Official Review · Reviewer_BAFm · 2022-10-27

**Confidence:** 4
**Correctness:** 3
**Technical Novelty And Significance:** 2
**Empirical Novelty And Significance:** 3
**Recommendation:** 5

**Clarity, Quality, Novelty And Reproducibility:**

Generally, the writing is relatively clear and follows a logical structure. However, the authors should demonstrate the superiority of the proposed method and why we should use the MLP-Mixer for the graph classification task. Just generalizing the MLP-Mixer from images to graphs makes the novelty limited. The writing and experimental settings are clear, which may help reproduce.


**Strength And Weaknesses:**

Strength:
- This paper generalizes MLP-Mixer from images to graphs and demonstrates the improvement in performance.
- The designs of Graph MLP-Mixer are reasonable. And the authors show the differences between MLP-Mixer for images and graphs
- This paper is well written and easy to follow. Detailed descriptions of the background are provided to make the paper more understandable.

Weakness:
- The contributions of this paper are limited. The authors simply move the MLP-mixer from CV domain to graph domain without explaining why this generalization is required for graphs.
- The motivation of this paper is that GNNs suffer from over-squashing and poor long-range dependencies. But these two issues mainly occur in large graphs. For graph classification task, the graphs are usually small. Can the authors provide some experimental results to demonstrate these two problems?
- The Positional Encoding and drop edges are  important for Graph MLP-mixer. How about just add these two component into the base models, such as GCN and GatedGCN?
- The performance is not comparable to the state-of-the-art models, for example, GNN-AK[1] , Graph transformers and other models on  the OGB Leaderboards. Although the performance is not necessary to be very good, the authors should demonstrate the superiority of the Graph MLP-Mixer than other models. For MLP-Mixer in CV domain, it can achieve comparable performance with CNNs and ViT.

[1] Zhao, Lingxiao, et al. "From stars to subgraphs: Uplifting any GNN with local structure awareness." arXiv preprint arXiv:2110.03753 (2021).


**Summary Of The Paper:**

This paper proposes a Graph MLP-Mixer model that generalizes the MLP-Mixer from images to graphs. Specifically, it first adopts graph clustering algorithms to split the graph into overlapping patches. And then different GNNs are utilized to get patch embeddings. To encode the positional information in graphs, Node PE and Patch PE are used.  Experimental results show the proposed method can outperform the base GNN encoder.

**Summary Of The Review:**

This paper proposes a Graph MLP-Mixer model that generalizes the MLP-Mixer from images to graphs. The technical details are reasonable. But the authors should demonstrate the superiority of the proposed method and why we should use the MLP-Mixer for the graph classification task.

---

> ### Author Response · Authors · 2022-11-15
> **Response to Reviewer BAFm (Part 1/3)**
>
> We thank the reviewer for her/his careful reading and comments regarding our work. Please, see below our answer to the raised comments/questions and find the revisions in blue color in the rebuttal manuscript.
>
> ---
>
> > **Reviewer**: The contributions of this paper are limited. The authors simply move the MLP-mixer from CV domain to graph domain without explaining why this generalization is required for graphs.
>
> **Authors**: The major contribution and main novelty of this paper is to design a GNN architecture that is not limited by poor long-range dependencies while keeping a low computational complexity. We described in Section 1 the standard MP-GNNs and their limitations in terms of isomorphism expressivity that can be improved with WL-GNNs and graph positional encoding. However, none of these expressive networks are able to deal properly with long-distance communication between nodes. The standard approach to propagate information between far away nodes is to stack several layers, given that the support of message-passing functions increases by one-hop at each layer. However, increasing the number of layers results in an exponential growth of the receptive field of the nodes. Hence, a large volume of neighbors (with their vectorial representations) are squeezed into one single vector. Such long-range dependencies and fast volume growth lead to the problem of over-squashing for all GNN architectures. To alleviate this problem, similarly to RNNs, Graph Transformers leverage the full-attention mechanism to address over-squashing but also significantly increase the complexity from linear O(E) to quadratic O(N^2) (E is the #edges, N is the #nodes), resulting in a computational bottleneck [1,2]. In order to overcome over-squashing for GNNs, we leverage the recent MLP-Mixer architecture that solves the issue of long-distance dependencies while keeping a low complexity. The above reasons justify the motivation to generalize ViT/MLP-Mixer to graphs – this generalization is required to overcome over-squashing, one of the major limitations with current GNN design. Finally, generalizing GNNs to ViT/MLP–Mixer from computer vision is not straightforward as we detailed in Section 2. Challenges are basically grouped into four generalization questions; How to extract graph patches? How to encode sub-graphs into vectors? How to preserve positional positions of nodes and patches? And how to reduce over-fitting?
>
> ---
>
> > **Reviewer**: The motivation of this paper is that GNNs suffer from over-squashing and poor long-range dependencies. But these two issues mainly occur in large graphs. For graph classification task, the graphs are usually small. Can the authors provide some experimental results to demonstrate these two problems?
>
> **Authors**: We have provided additional experiments with the recent 2022 NeurIPS Long Range Graph Benchmark (LRGB) [3] to demonstrate that Graph MLP-Mixer is able to capture long-range interactions. In LRGB, Peptides-func and Peptides-struct are two graph-level prediction datasets, consisting of 15,535 graphs with a total of 2.3 million nodes. The graphs are one order of magnitude larger than ZINC, MolTOX21, MolHIV and MolPCBA with 151 nodes per graph on average and a mean graph diameter of 57. As such, they are better suited to evaluate models enabled with long-range dependencies, as they contain larger graphs and more data points. The performance is reported in Table 2 and Table 5 in the rebuttal revision version and in Table 15 in the Appendix. We summarize the main results below.
>
> (1) Graph MLP-Mixer achieves SOTA with the best scores of 0.6846 on Peptides-func and 0.2478 on Peptides-struct (Table 5), demonstrating the ability of the model to better capture long-range relationships.
>
> (2) Compared with MP-GNNs (Table 2), Graph MLP-Mixer significantly outperforms the base MP-GNNs; we can observe an average 0.030 Average Precision improvement on Peptides-func and an average 0.030 MAE decrease on Peptides-struct.
>
> (3) Graph MLP-Mixer offers a low-cost architecture compared to the related architectures, GNN-AK+ [4] and SUN [5]. For example, SUN [5] gives similar performance to Graph MLP-Mixer, 0.6730 on Peptides-func and 0.2498 on Peptides-struct, but requires 30x memory and 18x training time (Table 15). Graph MLP-Mixer is cost-efficient and scalable with large graphs.
>
> ---

---

> > ### Author Response · Authors · 2022-11-15
> > **Response to Reviewer BAFm (Part 2/3)**
> >
> > > **Reviewer**: The Positional Encoding and drop edges are important for Graph MLP-mixer. How about just add these two components into the base models, such as GCN and GatedGCN?
> >
> > **Authors**: **Positional Encoding (PE)**: We have already augmented all the base models (GCN, GatedGCN, GINE and GraphTrans) with the same type of PE as Graph MLP-Mixer to ensure a fair comparison. For example, we use random walk structural encoding (RWSE) for molecular data in Graph MLP-Mixer and all base GNNs. As the most appropriate choice of PE is dataset and task dependent, we follow the practice from [6]. Details about the use of PE are reported in Table 7 and Table 8 in the Appendix.
> >
> > **Drop edges**: First, let us observe that the drop edge technique we use in the model is different to the standard data augmentation techniques such as DropEdge [7], and G-Mixup [8], which either add slightly modified copies of existing data or generate synthetic based on existing data. Our dropping edge mechanism is different and actually specific to the Graph MLP-Mixer model. Its utility is to produce different graph patches at each epoch because METIS, the graph partition algorithm we use, is deterministic and produces the same node partitions at each run. We propose to introduce some perturbations in METIS as follows. Let  G=(V, E) be the original graph and G’=(V, E’) be the graph after randomly dropping a small set of edges. We apply METIS graph partition algorithm on G’ to get slightly different node partitions {V_1, …, V_P}. Then, we extract the graph patches {G_1, …, G_P} where Gi = (V_i, E_i) is the induced subgraph of the original graph G, and not the modified G’. This way, we can produce distinct graph patches at each epoch that retain all the nodes and edges from the original graph. Finally, we observe that this patch augmentation technique is independent of the GNN encoder s.a. GCN, GAT, GIN or GatedGCN.
> >
> > In summary, the performance gain over base models is not due to either the use of additional PE or drop edges.
> >
> > ---
> >
> > > **Reviewer**: The performance is not comparable to the state-of-the-art models, for example, GNN-AK[4] , Graph transformers and other models on the OGB Leaderboards. Although the performance is not necessary to be very good, the authors should demonstrate the superiority of the Graph MLP-Mixer than other models. For MLP-Mixer in CV domain, it can achieve comparable performance with CNNs and ViT.
> >
> > **Authors**: Graph MLP-Mixer is competitive to SOTA models.
> >
> > We improved our model by applying alternatingly Patch Encoder and Mixer Layer and we call this updated model Graph MLP-Mixer*. The original Graph MLP-Mixer was sequential; the Patch Encoder was first applied to embed graph patches into a vectorial representation (no further update was possible) and then the Mixer Layer combined the patch representations. The updated version improves the node and the patch representations alternatively, which results in better performance as more information is exchanged between the nodes and the patches. Other components of the proposed architecture remain the same. A detailed description of the improved model can be found in Appendix A.9 and the results are reported  in Table 5, Table 14 and 15.
> >
> > Specifically, for small molecular graphs, Graph MLP-Mixer* achieved 0.0794 on ZINC, and 0.7912 on MolHIV, achieving comparable performance of SOTA models (ZINC: 0.070 (GPS) 0.084 (SUN); MolHIV: 0.7880 (GPS), 0.8003 (SUN)).
> >
> > For larger molecular graphs, Graph MLP-Mixer* showed superiority compared to both MP-GNNs and SOTA models on Peptides-func and Peptides-struct, two graph-level prediction datasets from Long Range Graph Benchmark (LRGB) [3] we discussed above.
> >
> > Besides, Graph MLP-Mixer offers better complexity and scalability. Experimentally, we observed that most Graph Transformer (GT) models need more training time (Table 14: GT models SAN and GrapTrans requires 843sec/epoch and 360sec/epoch respectively vs. 222sec/epoch for Graph MLP-Mixer).  As for some expressive GNNs such as the recent 2022 ICLR GNN-AK [4] mentioned by the reviewer and the 2022 NeurIPS SUN [5], they are limited by the huge memory and training time consumption, especially on larger graphs. As shown in Table 15, when training on datasets with hundreds of nodes, GNN-AK+ and SUN require 15x and 30x memory respectively compared to Graph MLP-Mixer. It indicates that Graph MLP-Mixer is more favorable in terms of scalability.
> >
> > ---

---

> > > ### Author Response · Authors · 2022-11-15
> > > **Response to Reviewer BAFm (Part 3/3)**
> > >
> > > > **Reviewer**: However, the authors should demonstrate the superiority of the proposed method and why we should use the MLP-Mixer for the graph classification task. Just generalizing the MLP-Mixer from images to graphs makes the novelty limited.
> > >
> > > **Authors**: We hope to have addressed above most of the questions raised by the reviewer. Please, let us know if you need more clarification.
> > >
> > > ---
> > >
> > >
> > >
> > > References:
> > >
> > > [1] Kreuzer, Beaini, Hamilton, Letourneau, Tossou, Rethinking Graph Transformers with Spectral Attention, NeurIPS 2021
> > >
> > > [2] Ying, Cai, Luo, Zheng, Ke, He, Shen, Liu, Do Transformers Really Perform Bad for Graph Representation?, NeurIPS 2021
> > >
> > > [3] Dwivedi et al, Long range graph benchmark, NeurIPS 2022
> > >
> > > [4] Zhao et al, From stars to subgraphs: Uplifting any GNN with local structure awareness, ICLR 2022
> > >
> > > [5] Frasca et al, Understanding and Extending Subgraph GNNs by Rethinking Their Symmetries, NeurIPS 2022
> > >
> > > [6] Rampášek et al, Recipe for a general, powerful, scalable graph transformer, NeurIPS 2022
> > >
> > > [7] Yu et a, Dropedge: Towards deep graph convolutional networks on node classification, 2019
> > >
> > > [8] Han et al, G-Mixup: Graph Data Augmentation for Graph Classification, 2022

---

> > > > ### Author Response · Authors · 2022-11-19
> > > > **Follow-up on Rebuttal Feedback**
> > > >
> > > > Dear Reviewer BAFm,
> > > > &nbsp;
> > > >
> > > > We thank you again for taking your time reviewing our work. We put our best effort to answer your questions.
> > > > &nbsp;
> > > >
> > > > We would appreciate to know if you need further clarification or have further question.
> > > > &nbsp;
> > > >
> > > > If you are satisfied with our answers, please consider revising your score.
> > > > &nbsp;
> > > >
> > > > Best regards,
> > > >
> > > > The Authors

---

### Author Response · Authors · 2022-11-18
**Engagement Appreciated**

Dear Reviewers,
&nbsp;

The discussion period is closing tomorrow.
&nbsp;

We would like to thank you again for your time reading and evaluating our work.
&nbsp;

Your questions and comments helped us to improve greatly the paper quality -- We have revised and uploaded a new version of the manuscript, with the revisions in blue color.
&nbsp;

We have also answered thoroughly your questions individually and point-by-point, with supplementary experiments as suggested.
&nbsp;

We would appreciate to know if you have any additional questions, concerns or clarifications you would like to ask us.
&nbsp;


If we have addressed your concerns, please, consider revising your score.
&nbsp;

We have put a lot of effort into this work, and we also tried to write the best possible rebuttal to answer your questions.
&nbsp;

Thank you very much for your understanding.
&nbsp;

With best regards,

The Authors

---

### Author Response · Authors · 2022-11-21
**Summary of the revised manuscript**

Dear Reviewers,
&nbsp;

Thank you again very much for your time.
&nbsp;

We summarize below the main contribution of the paper as well as the new experiments.
&nbsp;

**Main contribution:** To the best of our knowledge, the proposed work is the first GNN architecture that simultaneously 1) be expressive in terms of graph isomorphism (the model can distinguish at least 3-WL SR25 isomorphic graphs), 2) captures long-range dependencies (SOTA performances on the 2022 NeurIPS long-range LRGB dataset [1] and artificial TreeNeighbour dataset [2]), and 3) offers memory and speed efficiency (our model surpasses the 2022 ICLR GNN-AK+ [3] and 2022 NeurIPS SUN [4]). We believe this class of GNNs will spur a new line of research with more emphasis on the trade-off between long-range dependencies and low computational complexity, hence going beyond standard message-passing GNNs.
&nbsp;

**New experiments:**
+ Experiments on LRGB [1] to evaluate the property of long-range dependencies of the proposed model and comparison with the state-of-the-art including the most related models with the 2022 NeurIPS SUN and GSP [4,5] and the 2022 ICLR GNN-AK+ [3].
+ Experiments on ZINC, MolHIV and MolPCBA with an upgraded model GNN-MLP-Mixer*, which performs comparably to the best models on ZINC and MolHIV.
+ Experiments on the synthetic SR25 dataset with at least 3-WL isomorphic graphs [6]. Our model is the only model that can distinguish these non-isomorphic graphs with a low linear complexity (compared to GNN-AK+ [3]).
+ Experiments on the artificial TreeNeighbour to demonstrate that over-squashing can lead to both under-fitting and over-fitting. Besides, our model has the best generalization performance.
+ Experiments to investigate METIS vs. random partitioning, the effects of positional encoding and the k-hop size extension on the long-range Peptides-func datasets.
&nbsp;

We hope that the revision of the manuscript with the new experiments and our point-to-point answers have clarified most of the reviewers’ concerns. We are happy to answer any additional questions to clarify further.
&nbsp;

With best regards

The Authors
&nbsp;


References

[1] Dwivedi et al, Long range graph benchmark, NeurIPS 2022

[2] Alon, Yahav, On the Bottleneck of Graph Neural Networks and its Practical Implications, ICLR 2021

[3] Zhao et al, From stars to subgraphs: Uplifting any GNN with local structure awareness, ICLR 2022

[4] Frasca et al, Understanding and Extending Subgraph GNNs by Rethinking Their Symmetries, NeurIPS 2022

[5] Rampášek et al, Recipe for a general, powerful, scalable graph transformer, NeurIPS 2022

[6] Balcilar et al, Breaking the limits of message passing graph neural networks, ICML, 2021

---

### Decision · Program_Chairs · 2023-01-20

**Decision:**

Reject

**Justification For Why Not Higher Score:**

The updated version of the paper during the rebuttal helped address some concerns around the method's performance for tasks that require capturing long-range dependencies, but the authors needed to make significant changes to their core model (introduced in the appendix), which represent a strong departure from both the original model and story. A more careful revision of the paper is required to integrate these changes and make the story more coherent again.

**Justification For Why Not Lower Score:**

N/A

**Metareview: Summary, Strengths And Weaknesses:**

This paper proposes a novel GNN architecture for graph-level prediction tasks. The model (Graph MLP-Mixer) takes inspiration from a recent computer vision model architecture (MLP Mixer) and proposes a scheme, where the input graph is first split into overlapping clusters (“patches”) using a standard graph clustering algorithm, which is afterwards processed using multiple message passing stages, combining both typical GNN-style message passing on each individual patch, and MLP-Mixer inspired message passing across patches. This architecture is validated on a range of graph-level classification and regression tasks, showing competitive performance with prior methods.

The reviewers have highlighted that this paper is overall well-written and well-organized. The proposed method is novel and an interesting generalization of aspects of the MLP Mixer architecture to graphs. The reviewers further highlight the extensive experimental evaluation which highlights the competitive performance of the method.

Concerns raised by the reviewers were that the method is not competitive on all tasks, especially when considering state-of-the-art models such as GNN-AK [1], and that the main claims around over-squashing and long-range dependencies were not adequately validated. Given the large number of GNN architecture variants available in this domain, the concern was that authors would have to show convincing evidence that this type of generalization (MLP Mixer from the image domain to the graph domain) is required for graphs or provides sufficient benefits to warrant the added complexity of the method.

The authors have responded to these concerns by adding additional benchmark tasks aimed at evaluating long-range dependencies, as well as introducing a new model variant, termed MLP-Mixer* which uses a new iterative message passing scheme to update patch-level representations.

In the discussion call (and after carefully reading the paper myself), it became clear that this new model variant (introduced during the rebuttal phase) presents a significant departure from the original method and the updated paper significantly declined in clarity, as the paper now introduces two major model variants with unclear trade offs between the two; on some of the tasks, only the new model variant is compared, whereas on others both the old and the new model variant are compared. Similarly, it appears that the new model variant does well when compared to a simpler ViT-variant of the model, while the old variant does not, which raises concerns around the overall necessity of the modeling choices (see Tables 3 and 4). The definition of the new model variant is deferred to the appendix, significantly hurting clarity of the paper. Reviewer BAFm further expressed the concern that the original story around efficiency benefits of the model will no longer hold with the newly introduced variant and that a revision of the paper is required to address these concerns.

Overall, I agree with these concerns and think that this paper requires a significant revision in order to clear up the story, model presentation, and experimental section. I recommend that directly introduce their new, improved model (which was introduced in the rebuttal) in a revised version of the paper, along with new appropriate baseline comparisons (e.g. a ViT version of the model that includes all the improvements of the new MLP-Mixer* model variant) and ablations. I very much believe that this would make the paper significantly more valuable for the community and increase its impact.

[1] Zhao et al., From stars to subgraphs: Uplifting any GNN with local structure awareness (2021).


**Summary Of Ac-Reviewer Meeting:**

I was able to meet with 3 of the 4 reviewers; reviewer oFvZ was unable to attend the meeting due to time zone constraints.

Reviewer jedP argued that originally they were leaning towards rejection as the paper did not have important experiments to demonstrate effectiveness — specifically no long-range dependency experiments. The authors in response looked at long-range graph benchmark and introduced an extension of their model termed Graph MLP-Mixer*. This is a **significant** change to their model. Quality of the paper is somewhat compromised; no significant experiments on the long-range graph benchmark – only final performance, no deeper analysis done. Overall, very marginally leaning towards accept, but would not champion the paper.

Reviewer Fxzu argued that authors have addressed most of their original concerns in the rebuttal. Remaining concern: Evaluation does not include dataset in OGB dataset comparing to SoTA. Reasons for acceptance: 1) Novel contribution around patching graph and doing the classification in a hierarchical fashion, 2) Evaluation results look good, 3) Main idea of paper is pretty interesting. However, quality of paper is compromised in revision: many points are covered in appendix but not in main paper. How model captures long-range dependencies is not well analyzed – there should be some qualitative analysis. Authors could address this by down-weighing their claims around long-range dependencies and oversquashing.

Reviewer BAFm positively highlighted that for the original model (before the rebuttal) performance was comparable (to prior work) while efficiency was better. For new model variant (Graph MLP-Mixer*) introduced during rebuttal, original claim around efficiency will not hold; will not align well with the new model (even though performance is now better). Overall, will need a big revision to make the story coherent.

Summary:

I am weighing the concerns around compromised quality/clarity of the paper, due to the significant changes that the authors had to introduce during the rebuttal to address initial reviewer concerns, as very high. I fully agree with the reviewers that the paper would heavily benefit from being revised to clear up the story after all the newly introduced changes to method and experiments, and therefore recommend to reject the paper in its current form.